# Permuton-induced Chinese Restaurant Process

**Masahiro Nakano, Yasuhiro Fujiwara, Akisato Kimura, Takeshi Yamada, Naonori Ueda**
NTT Communication Science Laboratories, NTT Corporation
{masahiro.nakano.pr, yasuhiro.fujiwara.kh, akisato.kimura.xn,
takeshi.yamada.bc, naonori.ueda.fr}@hco.ntt.co.jp

## Abstract

This paper proposes the permuton-induced Chinese restaurant process (PCRP), a stochastic process on rectangular partitioning of a matrix. This distribution is suitable for use as a prior distribution in Bayesian nonparametric relational model to find hidden clusters in matrices and network data. Our main contribution is to introduce the notion of permutons into the well-known Chinese restaurant process (CRP) for sequence partitioning: a permuton is a probability measure on $[0, 1] \times [0, 1]$ and can be regarded as a geometric interpretation of the scaling limit of permutations. Specifically, we extend the model that the table order of CRPs has a random geometric arrangement on $[0, 1] \times [0, 1]$ drawn from the permuton. By analogy with the relationship between the stick-breaking process (SBP) and CRP for the infinite mixture model of a sequence, this model can be regarded as a multi-dimensional extension of CRP paired with the block-breaking process (BBP), which has been recently proposed as a multi-dimensional extension of SBP. While BBP always has an infinite number of redundant intermediate variables, PCRP can be composed of varying size intermediate variables in a data-driven manner depending on the size and quality of the observation data. Experiments show that PCRP can improve the prediction performance in relational data analysis by reducing the local optima and slow mixing problems compared with the conventional BNP models because the local transitions of PCRP in Markov chain Monte Carlo inference are more flexible than the previous models.

## 1 Introduction

The multi-dimensional extension of the Chinese restaurant process (CRP) [9, 50] for infinitely exchangeable random arrays is one of the most significant unsolved issues in the field of Bayesian nonparametrics and its applications to machine learning. The standard CRP is the stochastic process representing random clustering of a sequence using the metaphor of customers sitting at tables. Typically, it is often used when we need to cluster a sequence where the number of clusters is unknown in advance. The magic of CRP is that it requires only a finite number of parameters for finite data; nevertheless, the model itself has infinite representational power. Specifically, it is known that a mixture model using CRP is theoretically equivalent to the Dirichlet process mixture model (DPMM) [51, 27] represented by the stick-breaking process (SBP) [57] with an infinite number of parameters. The relationship between the two DPMM representations, CRP and SBP, can be explained by whether the intermediate random variables in de Finetti's representation theorem [28, 37] are marginalized out or not (Figure 1, (a) and (b)). CRP has contributed greatly to the development of the BNP machine learning field in the sense that it has made BNP not only theoretical but also practical. It enables models with infinite representational power to be accurately simulated on a computer with a finite number of parameters without any approximation (e.g., finite truncation). In fact, a variety of extensions of CRP have been studied, including the nested CRP [16, 17, 60], the Chinese restaurant franchise [61], the tree-structured CRP [6], the distance-dependent CRP [14, 15, 31]. However, in the development of CRP, the extension to BNP relational models has remained a historical conundrum.

35th Conference on Neural Information Processing Systems (NeurIPS 2021).

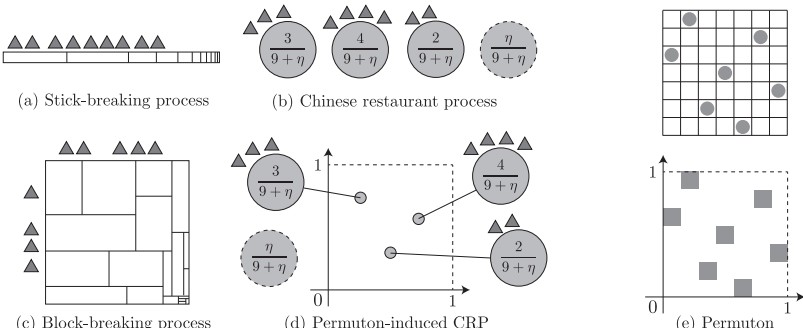

Figure 1: (a) SBP representation for sequence partitioning based on de Finetti's representation. (b) CRP representation for sequence partitioning. Each customer (triangle) sequentially choose a table (dark circle). (c) BBP representation for rectangular partitioning based on AHK representation theorem. (d) Our proposed PCRP. We recall that a certain class of rectangular partitioning has a bijection to a certain class of permutation. Therefore, if we could add to the order based on permutations for CRP tables, we could convert the table assignments of data by CRP into rectangular partitioning. To achieve this, we introduce a random geometric location on $[0,1] \times [0,1]$ into the CRP table drawn from permuton. (e) Illustration of permuton. Top shows an array representation of a permutation $\sigma = 5724163$. The horizontally $i$-th circle has the height $\sigma(i)$. Bottom shows the corresponding permuton to $\sigma = 5724163$, i.e., the probability measure on $[0,1] \times [0,1]$.

As a special case of the BNP relational models, the infinite relational model (IRM) [38] consisting of the product of CRPs was proposed. The IRM can express random rectangular partitioning of a matrix; however, its representational capability is restricted to a minimal subset of rectangular partitioning called *regular grid* (Figure 2, left). Therefore, in order to obtain a model with higher representational capability, a variety of BNP models for relational data have been rapidly developed in recent years [38, 55, 54, 20, 53, 58, 46, 36, 19, 7, 23, 26, 43, 44, 30, 21, 25, 24, 56]. Recent excellent surveys can be found in [22, 49]. In particular, the Mondrian process (MP) [55, 54] and the rectangular tiling process (RTP) [48] have been proposed as new BNP models to expand the expressive classes of rectangular partitioning. However, both of them require an infinite number of parameters as intermediate random variables in the Aldous-Hoover-Kallenberg (AHK) representation theorem [8, 34, 37] (an extension of de Finetti's theorem to multi-dimensional arrays), similar to the SBP representation for DPMM. As a result, Bayesian inference to those models always had to take care of an infinite number of parameters, which compromise the flexibility of local movements for the Markov chain Monte Carlo (MCMC) inference and make it highly affected by local optima and slow mixing problems. As a natural question, we wonder if it is possible to marginalize out the infinite number of intermediate random variables of these models, as in the CRP representation of DPMM. Unfortunately, an approach to achieve this has remained unsolved for a long time.

This paper deals with the multi-dimensional extension of CRP to represent wider classes of rectangular partitioning than regular grid partitioning. As an important stepping stone for this, the block-breaking process (BBP) [47] has been proposed very recently as a multi-dimensional extension of SBP (Figure 1, (c)). The key insight for the BBP construction is that rectangular partitioning can be represented indirectly using *permutations*. Specifically, a special class of permutations called *Baxter permutation* [13], which has a one-to-one correspondence with a particular class of rectangular partitioning, is used to extend the stick-breaking procedures of SBP for a sequence to the block-breaking procedures for a matrix (Figure 3). Although BBP can indeed be used as an infinitely exchangeable relational model, it requires an infinite number of redundant parameters, similar to SBP, making Bayesian inference highly affected by local optima and slow mixing problems. Therefore, a CRP-like representation that avoids the infinite number of redundant parameters of BBP, analogous to the relationship between SBP and CRP for DPMM, has been desired. Our contributions are summarized as follows:

**Construction of new stochastic process -** Section 3.1 proposes a stochastic process such that each table of the CRP has a random coordinates on $[0,1] \times [0,1]$. Specifically, we suppose that the coordinates of each table is independent and identically distributed (i.i.d.) random variable drawn from a *permuton* [35] on $[0,1] \times [0,1]$. Permuton is a probabilistic measure on $[0,1] \times [0,1]$ and can also be regarded as a prior model for random permutations. By exploiting the correspondence between permutations and rectangular partitions [62, 52], PCRP can be used as a generative probabilistic

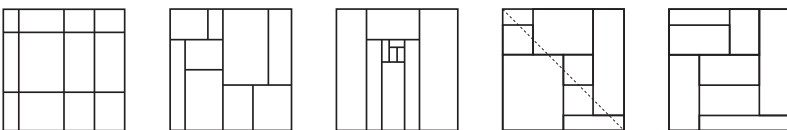

Figure 2: Several classes of rectangular partitions appear in this paper. The details of each class will also be described in the main body of Section 2.1. **Left:** Regular grid partitioning. Each block is characterized by the product of the row and column clusters. **Second from left:** Hierarchical partitioning. **Third from left:** Hierarchical partitioning with one place deeper in each layer. **Fourth from left:** Diagonal rectangulation. **Right:** Generic rectangulation.

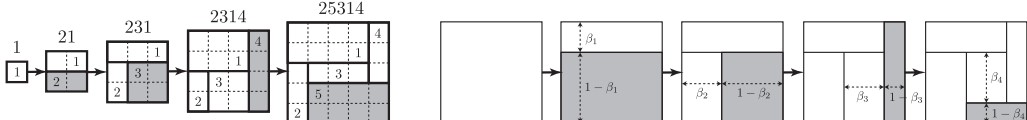

Figure 3: Illustration of BBP [47], which can be regarded as a multi-dimensional extension of SBP. First BBP uses a Markov process on Baxter permutations, and transforms the evolution of Baxter permutations to the evolution of floorplan partitioning (i.e., rectangular partitioning ignoring the size of each rectangle block) by using the one-to-one correspondence between Baxter permutations and floorplan partitioning [33] (**Left**). Next BBP uses the block-breaking procedures to sequentially assign the random size to each rooms floorplan partitioning (**Right**).

model of rectangular partitioning of a matrix. While BBP [47] (i.e., a multi-dimensional extension of SBP) always has an infinite number of redundant intermediate variables, our PCRP can be composed of varying size (finite) intermediate variables in a data-driven manner depending on the size and quality of the observation data. As a result, PCRP can lead to very flexible local movements in Bayesian inference, reducing the problems of local optima and slow mixing.

**Unified framework for various classes of rectangular partitioning -** Conventionally, the stochastic processes for each class of rectangular partitioning have been constructed individually. However, according to the design of the permuton, PCRP provides a unified framework for various classes of rectangular partitioning. Section 3.2 shows an approach to tune the permuton of PCRP to adapt it to various classes of rectangular partitioning.

**New representation of BNP model -** As mentioned earlier, a multi-dimensional extension of CRP is a historical conundrum, and thus PCRP does not provide a perfect solution to this issue. In Section 4, the theoretical imperfection of PCRP is clarified. Specifically, we intuitively explain that projectivity and exchangeability of PCRP change when viewed from two different perspectives: rectangular partitioning of a matrix and table assignment of data. Thus, we propose a new representation of the BNP model as a way to overcome this imperfection, that applies PCRP as a bridging state to existing infinitely exchangeable relational models, inspired by the MCMC inference via bridging [42].

## 2 Preliminaries

### 2.1 Permutation, permutation class and its relation to rectangular partitioning

**Permutation and rectangulation -** A permutation of a set $S$ is defined as a bijection from $S$ to itself. Specifically, this paper always focus on permutations of natural numbers $S = [n] := \{1, 2, \ldots, n\}$ ($n \in \mathbb{N}$). For convenience, we use one-line notation for a permutation $\sigma = \sigma_1 \sigma_2 \ldots \sigma_n$. For example, the permutation $\sigma = 312$ means $\sigma_1 = 3, \sigma_2 = 1, \sigma_3 = 2$. A rectangulation (also called rectangular partitioning) is a tiling of a rectangle by a finite number of disjoint rectangles such that no four of its rectangles share a single corner. In recent years, it has become clear that there is a close relationship between permutations and rectangulations. Here we focus in particular on the fact that every permutation can be uniquely converted into a *generic* rectangulation (Figure 2, right), that is, the class of rectangulations without any constraints:

**Proposition 2.1** *(See Proposition* 6.2 *in [39] and Proposition* 4.2 *in [52]) There is a surjective map from permutations to generic rectangulations.*

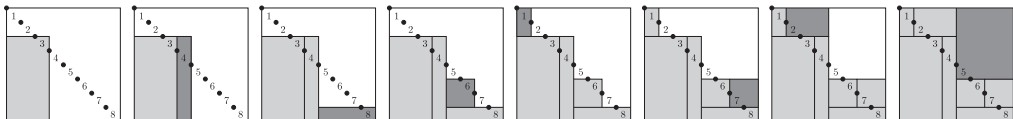

Figure 4: Map from permutations (e.g., $\sigma = 34861725$) to diagonal rectangulations [39]. We first draw distinct *diagonal points* on the diagonal, with one of the points being the top-left corner and another being the bottom-right corner. We perform the following steps sequentially. Let $R$ be the union of the rectangles drawn in the first $i-1$ steps. To draw the $i$th rectangle, we consider the label $i$ on the diagonal. If the diagonal point $p$ on the diagonal immediately above or left of the label $i$ is not in $R$, then the upper left corner of the new rectangle is the rightmost point of $R$ immediately to the left of $p$. If the diagonal point $p$ immediately below or right of the label $i$ is not in $R$, then the lower right corner of the new rectangle is the highest point of $R$ immediately below $p$. If $p$ is in $R$, then the lower-right corner of the new rectangle is the rightmost point of $R$ immediately to the right of $p$.

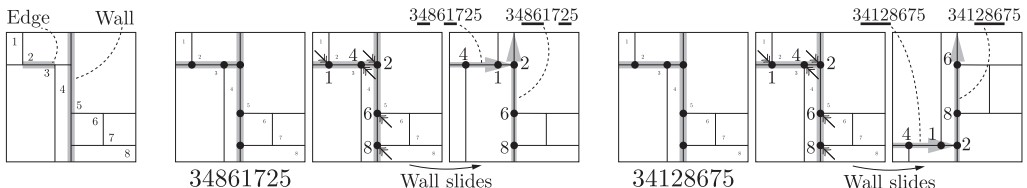

Figure 5: Transformation from diagonal rectangulations to generic rectangulations [52]. **Left**: Two notions, *edge* and *wall*. An *edge* of the rectangulation is a line segment contained in the side of some rectangles such that the endpoints of the line segment are vertices and the segment has no vertices in its interior. A *wall* of the rectangulation is a maximal union of edges forming a line segment. **Middle and right**: Two examples of mapping from permutations (e.g., $\sigma = 34861725$ and $\sigma = 34128675$) to generic rectangulations. First, we convert the permutation to a diagonal rectangulation. Then we assign to the vertex on the *walls* the label of the block that contains that vertex as its own upper left or lower right corner. Finally, the order of the vertices on the wall will be rearranged according to the permutation. Specifically, vertices on the horizontal wall are aligned in permutation order from left to right, and vertices on the vertical wall are aligned in permutation order from bottom to top.

Fortunately, we can constructively obtain this map as the following two-stage transformation from a permutation to a general rectangulation by way of a *diagonal* rectangulation. A rectangulation is *diagonal* if the interior of each rectangle has an area that intersects the diagonal. Figure 4 and Figure 5 describe the details of the first and second steps, respectively. We use this transformation as an important component in our generative model of PCRP (in Section 3.1).

**Permutation classes and rectangulation classes -** To investigate the properties of the permutation, we can focus on subsets of permutations as *permutation classes*. To identify a permutation class, we introduce the notion of *occurrences*. We suppose that $\tau$ and $\sigma$ are permutations of size $k$ and $n$, respectively. The *occurrences* of pattern $\tau$ in $\sigma$ is a subsequence $\sigma_{i_1}, \ldots, \sigma_{i_k}$ that is *order-isomorphic* to $\tau$, that is, for all indices $s, t \in [k]$, we have $\sigma_{i_s} < \sigma_{i_t} \iff \tau_s < \tau_t$. One way to characterize a permutation class is often to focus on what patterns (typically very short permutations) are *not* contained as occurrences in the permutations belonging to that class. Conventionally, various permutation classes have been studied in depth, however, here we would like to focus on several permutation classes that are particularly relevant to rectangular partitioning. The details of each class will be explained in the supplementary material. It should be emphasized here that, surprisingly, permutations of a particular class have a one-to-one correspondence with rectangulations of a particular class. A brief list is given below. We will use this fact in Section 3.2 to show that PCRP can potentially serve as a unified model for the various classes of rectangular partitioning.

**2-clumped permutation**: The 2-clumped permutations have a bijection to *generic rectangulations* (Figure 2, right) [52], that no restrictions are required.

**Baxter permutation**: and **twisted Baxter permutation**: The Baxter permutations have a bijection to *diagonal rectangulations* (Figure 2, fourth from left) [62, 5, 32], which has a representative in which every rectangle's interior intersects the diagonal.

**Separable permutation**: Separable permutations have a bijection to *hierarchical and diagonal rectangulations* [59, 5], a subset of *hierarchical partitioning* (Figure 2, second from left), which are

expressed as binary trees where nodes represent a vertical or horizontal separation of a rectangle into two disjoint rectangles.

**Separable skew-merged permutation**: Separable skew-merged permutations have a bijection to a special subset of hierarchical partitioning, in particular, those where only one rectangle is allowed to be further cut in each layer (Figure 2, third from left) [45].

## 2.2 Permuton

In this paper, we introduce *permuton* [35] as a very useful tool for handling random permutations. Before defining permuton, we begin with a geometric interpretation of a permutation (Figure 1, (e) top). To a permutation $\sigma$, we can associate a probability measure $\gamma_\sigma$ on $[0, 1] \times [0, 1]$ as follows. We first divide $[0, 1] \times [0, 1]$ into an $n \times n$ grid of squares of size $1/n \times 1/n$. Then, we can define the density of $\gamma_\sigma$ on the square in the $i$-th row and $j$-th column to be the constant $n$ if $\sigma_i = j$ and $0$ otherwise. Therefore, $\gamma_\sigma$ can be regarded as a geometric representation of the permutation matrix of $\sigma$. More generally, we define a *permuton* to be a probability measure $\gamma$ on $[0, 1] \times [0, 1]$ with uniform marginals (Figure 1, (e) bottom): $\gamma([a, b] \times [0, 1]) = b - a = \gamma([0, 1] \times [a, b])$ for all $0 \le a \le b \le 1$. We emphasize that, for any permutation $\sigma$, the corresponding probability measure $\gamma_\sigma$ is a permuton. Specifically, the permutation $\sigma$ can be interpreted as the permuton $\gamma_\sigma$ given by the sum of Lebesgue area measures [35]: $\gamma_\sigma(A) = n \sum_{i=1}^{n} \mathrm{Leb}\big([(i-1)/n, i/n] \times [(\sigma(i)-1)/n, \sigma(i)/n] \cap A\big)$, for all Borel measurable set $A$ of $[0, 1] \times [0, 1]$, where $\mathrm{Leb}(\cdot)$ indicates the Lebesgue measure.

We will provide the intuition behind why permuton is useful in dealing with random permutations. We suppose that $\tau$ and $\sigma$ are permutations of size $k$ and $n$, respectively. We set $\widetilde{\mathrm{occ}}(\tau, \sigma) := (n!/k!(n-k)!) \cdot \#\{\text{occurrences of } \tau \text{ in } \sigma\}$. Intuitively, $\widetilde{\mathrm{occ}}(\tau, \sigma)$ means the probability to find a pattern $\tau$ in $\sigma$, when we take $k$ elements uniformly at random in $\sigma$. Similarly, if we replace permutation $\sigma$ with permuton $\gamma$, we can consider the probability of occurrence of a pattern $\tau$: $\widetilde{\mathrm{occ}}(\tau, \gamma) := \mathbb{P}[u_1, \ldots, u_k \text{ form a pattern } \tau]$, where $u_1, \ldots, u_k$ are i.i.d. points on $[0, 1] \times [0, 1]$ drawn from the permuton $\gamma$. Now, a natural question is whether these two $\widetilde{\mathrm{occ}}(\tau, \sigma)$ and $\widetilde{\mathrm{occ}}(\tau, \gamma)$ are generally consistent. The following theorem gives us an answer to this question:

**Theorem 2.2** *(See, e.g., Theorem 2.5 [29] and Lemma 3.5 [35]) We consider a sequence of permutations $\sigma_1, \sigma_2, \ldots, \sigma_n, \ldots$ of size tending to infinity $n \to \infty$. Then, we have $\gamma_{\sigma_n} \to \gamma \iff$ for every $\tau$, $\widetilde{\mathrm{occ}}(\tau, \sigma_n) \to \widetilde{\mathrm{occ}}(\tau, \gamma)$. Moreover, if $\tau$ and $\sigma$ are permutations of size $k$ and $n$, respectively, then $|\widetilde{\mathrm{occ}}(\tau, \sigma) - \widetilde{\mathrm{occ}}(\tau, \gamma_\sigma)| \le k(k-1)/2n$.*

As a result, random permutations belonging to some class can be controlled by the limit of sequences of permutons. We use the permuton as a prior model for the PCRP table coordinates (in Section 3.1).

## 3 Permuton-induced Chinese restaurant process (PCRP)

This section proposes a new stochastic process, which is our main contribution to this paper. First, we describe the stochastic process, PCRP, in terms of a generative model and show how we can introduce the notion of the permuton into CRP to obtain a model for rectangular partitioning. Next, we discuss how to design the permuton so that PCRP can represent some classes of rectangular partitioning.

### 3.1 Model description of PCRP from generative probabilistic point of view

PCRP is composed of an input matrix, the concentration parameter $\eta > 0$, and the permuton $\gamma$, i.e., a probability measure on $[0, 1] \times [0, 1]$. The specific design of the permuton will be discussed in the next subsection, thus the reader may continue to read assuming, for example, $\gamma(\cdot) = \mathrm{Leb}(\{\cdot\})$, that is, the uniform distribution on $[0, 1] \times [0, 1]$. The following four steps represent the overall picture of PCRP from the viewpoint of a probabilistic generative model.

**Step 1: Table assignment process of all row and column elements of the input matrix by a single CRP (Figure 6, first and second from left) -** Each row (white triangle) and each column (black triangle) of the input matrix chooses either an existing table to sit at with probability $S_t/(S+\eta)$ $(t = 1, \ldots, K)$ or a new table with probability $\eta/(S + \eta)$, as in a standard CRP [9], where $K$ is the number of tables, $S$ is the total number of customers seated at the tables, $S_t$ is the number of customers seated at the $t$-th table. If a new table is chosen, we draw the coordinates of the table

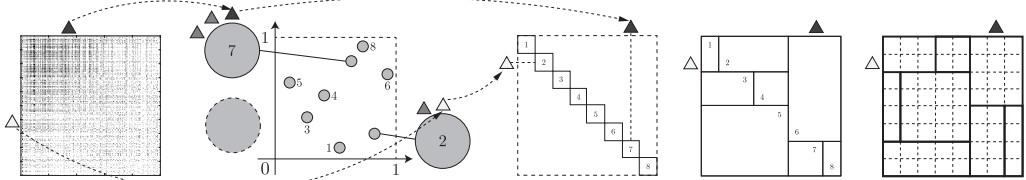

Figure 6: Overview of permuton-induced CRP, described in Section 3.1.

on $[0, 1] \times [0, 1]$ from the permuton $\gamma$. Here, for simplicity, we show a model that delivers row and column elements to a single CRP. However, the model can be easily extended to use different CRPs for rows and columns as in the conventional IRM [38] (e.g., using the Chinese restaurant franchise [61], where CRPs for rows and columns share common dishes).

**Step 2: Numbering the CRP tables and placement of customers on the sides of a rectangle (Figure 6, second and third from left)** - The CRP tables are numbered according to the vertical order of the coordinates, which leads to a permutation $\sigma = \sigma_1 \ldots \sigma_K$. Note that we recursively perform this numbering each time we add a new table. Next, we consider a rectangle (referred to as the outer rectangle) and its main diagonal and divide it into $K$ equal-sized line segments numbered $1, \ldots K$, from top left to bottom right. The customer corresponding to each row of the input matrix is moved to the left side position of the outer rectangle corresponding to the table number it belongs to. Similarly, we move the customer corresponding to each column of the input matrix to the top side position of the outer rectangle corresponding to the table number it belongs to.

**Step 3: Transforming permutation to generic rectangulation (Figure 6, fourth and fifth from left) -** We apply the map from permutations to rectangulations described in Proposition 2.1 (Figure 4 and 5) to the permutation $\sigma$ derived from the PCRP table coordinates in **Step 2**. This transformation is performed in two steps, first converting the PCRP table assignment (**Step 1**) to a diagonal rectangulation (Figure 6, fourth), and then converting it further to a generic rectangulation (Figure 6, right). As a result, PCRP yields a random rectangular partition of the input matrix.

**Intuitive interpretation of PCRP**: We recall that PCRP consists of two components: (1) random table assignments to all the row and column elements of the input matrix, and (2) random permutations represented by the table coordinates. The former table assignments serve to roughly group the rows and columns of the input data, and the latter random table coordinates serve to transform those rough groupings into detailed rectangular partitioning. The practical advantage of PCRP is that when a new table of CRPs is added, it can be inserted into every possible place of the rectangular partition (with the consistency that the whole is a rectangular partition). This allows us to eliminate the problem of the disadvantage of the conventional BBP, where the insertion of a new rectangle is restricted to a specific place in the lower right corner (Figure 3).

## 3.2 Design of permuton to provide various representational capabilities to PCRP

In this subsection, we discuss how to specifically choose a permuton $\gamma$ to restrict the rectangular partitions generated by PCRP to a specific class. We start with the simplest case of $\gamma(\cdot) = \mathrm{Leb}(\{\cdot\})$, called the *uniform permuton* [35]. As is well known, if $\sigma_n$ is a uniform random permutation of size $n$, then $\gamma_{\sigma_n}$ converges in distribution to this uniform permuton. We recall here that every permutation can be transformed into some rectangular partition. Thus it seems that PCRP with the uniform permuton can be used as a probabilistic model for arbitrary rectangular partitioning. However, there is one crucial point to note: The mapping between permutations (a set of all permutations) and generic rectangulations is not a one-to-one correspondence. This means that even if the random permutations are uniform, the corresponding random rectangular partitions they are transformed into are not uniform and have some bias. Namely, there are some biases in PCRP with the uniform permuton, such that some rectangular partitions are more likely to appear and some are less likely to appear. We could ignore this bias for practical use, but theoretically, it is essential to know how to eliminate it. We briefly introduce some permuton designs that are uniform for some class of rectangular partitioning. Here is a brief list, and details are given in the supplementary material.

**Hierarchical partitioning with one place deeper in each layer (Figure 2, third from left) -** Uniform partitioning of this class can be obtained by restricting the random permutation of coordinates in

the CRP tables to uniform separable skew-merged. If $\sigma_n$ is a uniform separable skew-merged permutation of size $n$, then $\gamma_{\sigma_n}$ converges in distribution to a deterministic permuton $\gamma$ (See, e.g., Theorem 3.3 and Section 3.2.1 [12]). Intuitively, the permuton $\gamma$ has an X-shaped density on $[0, 1] \times [0, 1]$.

**Hierarchical and diagonal rectangulation (Figure 2, second and fourth from left) -** Uniform hierarchical and diagonal rectangulations can be expressed by uniform separable permutations. If $\sigma_n$ is a uniform separable permutation of size $n$, then $\gamma_{\sigma_n}$ converges in distribution to the *Brownian separable permuton* [11, 10, 45].

**Diagonal rectangulation (Figure 2, fourth from left) -** Uniform diagonal rectangulations can be expressed by uniform Baxter permutations. If $\sigma_n$ is a uniform Baxter permutation of size $n$, then $\gamma_{\sigma_n}$ converges in distribution to the *Baxter permuton* [18].

**Generic rectangulation (Figure 2, right) -** To the best of our knowledge, there is still no way to construct a permuton that corresponds to 2-clumped permutations, which have a bijection to generic rectangulations. This means that it is currently tricky to explicitly design a clever permuton that will always restrict its samples to 2-clumped permutations. One possible strategy is to relax the permuton itself to one that can generate a broad class of permutations (e.g., $\gamma(\cdot) = \mathrm{Leb}(\{\cdot\})$), and instead use *rejection sampling* to restrict the generated permutations so that they are actually 2-clumped. Details are given in the supplementary material.

# 4 Intermediate level representation between SBP and CRP

PCRP would be very useful in the practical use of relational data analysis due to the following two advantages. (1) The model itself adjusts the model complexity in a data-driven manner according to the quality and size of the input data. (2) The design of permuton allows us to handle various classes of rectangular partitioning in a unified manner. In this section, we explore the former point more deeply from the viewpoint of the theory of Bayesian nonparametrics. A precise explanation would require the notions of *projective system*, *projectivity*, and *exchangeability*, which are far from the scope of this paper. Therefore, we would like to focus on an intuitive explanation while providing a rigorous discussion in the supplementary material.

**Motivation -** As well as the standard CRP for sequence partitioning being an infinitely exchangeable model, PCRP also has projectivity and exchangeability in terms of the CRP table assignments of data. However, from the perspective of rectangular partitioning model in relational data analysis, the projectivity and exchangeability of PCRP, unfortunately, do not hold. The implications of this fact will be explained in terms of the practical intuition for relational data analysis. We consider a pair of matrices $I \subset J \subset \{1, 2, \ldots, \} \times \{1, 2, \ldots, \}$. Namely, a smaller matrix $I$ is an extraction of some rows and columns of a larger matrix $J$. As it turns out, the only theoretical imperfection of PCRP can be explained as follows: the random rectangular partitioning of the smaller matrix $I$ that follows PCRP cannot be equivalent to the random rectangular partitioning of $I$ when the random rectangular partitioning of the larger matrix $J$ follows PCRP. In other words, PCRP does not have perfect self-similarity according to the size of the data in terms of the rectangular partitioning model, but only in the imaginary object of the CRP table assignments. This implies that PCRP may have to tune its model complexity (e.g., the concentration parameter of CRP) according to the size of the data. Nevertheless, in a practical sense, we believe that this is not such a significant issue. Rather, local optima and slow mixing problems in Bayesian inference have a more significant impact on the performance of the model. However, readers familiar with the theory of Bayesian nonparametrics may wish to have a BNP model that is strictly projective and exchangeable in terms of rectangular partitioning. For such readers, we also propose a new way to create an exact infinite exchangeable model while using PCRP. Specifically, we apply PCRP as a bridging state to existing infinitely exchangeable relational models, inspired by the MCMC inference via bridging [42].

We are interested in a set of rectangular partitioning of a matrix $I \subset \{1, 2, \ldots, \} \times \{1, 2, \ldots, \}$ whose rows and columns are indexed by natural numbers. We consider probability measures $\mu_X^I$ on $(\mathcal{X}_I, 2^{\mathcal{X}_I})$ where $\mathcal{X}_I$ is a set of rectangular partitioning of a matrix $I$ (i.e., a constrained combinatorial space), and hope to find an exact BNP model (Figure 7, left). Needless to say, a number of the various BNP models have actually been proposed as the exact BNP model $\mu_X^I$, including the IRM [38], the MP-based relational model [55], and the BBP-based relational model [47]. However, conventionally, these exact BNP models have significantly struggled with local optima and slow mixing problems in Bayesian inference. For example, in the standard MCMC inference phase, we wish to sample

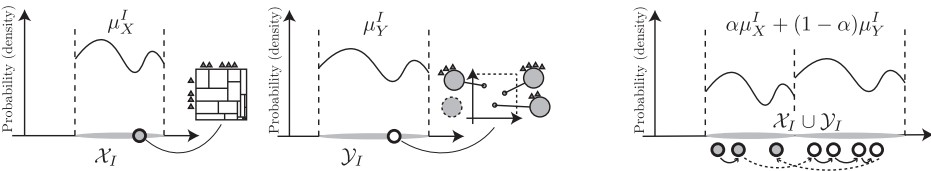

Figure 7: Illustration of exact BNP model with bridging states. **Left:** Existing exact BNP model on rectangular partitioning $\mathcal{X}_I$. **Middle:** PCRP on imaginary table assignments of data $\mathcal{Y}_I$. **Right:** BNP model on extended space $\mathcal{X}_I \cup \mathcal{Y}_I$. Sampling from $\mu_X^I$ is equivalent to drawing samples (gray circles) from $\mathcal{X}_I \cup \mathcal{Y}_I$ via the joint chain and discarding those (white circles) from $\mathcal{Y}_I$.

from the distribution $\mu_X^I$ over a constrained combinatorial space $\mathcal{X}_I$. Typically, by using some local moves, we can derive a Markov chain with a transition matrix, which may have slow mixing or even be non-ergodic. To mitigate these issues, we propose to introduce auxiliary spaces to relax the local movements in combinatorial spaces and to connect different regions of the sample space where communication is difficult or impossible.

**Exact BNP model with bridging states -** First, as the auxiliary space $\mathcal{Y}_I$, we introduce the table assignments of $I$ by CRP and a set of coordinates on $[0,1] \times [0,1]$ of the tables. The corresponding probability model $\mu_Y^I$ on $(\mathcal{Y}_I, 2^{\mathcal{Y}_I})$ should be PCRP described in the previous section (Figure 7, middle). Then, connecting the bridging state in the auxiliary space $\mathcal{Y}_I$ with samples in $\mathcal{X}_I$, we consider the combined probability model over the union space $\mathcal{X}_I \cup \mathcal{Y}_I$. To address probabilistic models on the union space $\mathcal{X}_I \cup \mathcal{Y}_I$, we modify the previous probabilistic models $\mu_X^I$ and $\mu_Y^I$ as follows: $\hat{\mu}_X^I(\cdot \in 2^{\mathcal{X}_I \cup \mathcal{Y}_I}) := \mu_X^I(\cdot \cap \mathcal{X}_I)$ and $\hat{\mu}_Y^I(\cdot \in 2^{\mathcal{X}_I \cup \mathcal{Y}_I}) := \mu_Y^I(\cdot \cap \mathcal{Y}_I)$. For simplicity of notation, the modified probabilistic models $\hat{\mu}_X^I$ and $\hat{\mu}_Y^I$ will be denoted as original $\mu_X^I$ and $\mu_Y^I$ in the following. Finally, we consider a probabilistic model $\mu_+^I$ (Figure 7, right) in form of $\mu_+^I = \alpha \mu_X^I + (1-\alpha)\mu_Y^I$, where $0 \le \alpha \le 1$ is a tunable real variable. Fortunately, we can treat this new model $\mu_+^I$ as a BNP model, that is, there uniquely exists a projective limit of $\mu_+^I$ ($I \to \mathbb{N} \times \mathbb{N}$) (See the supplementary materials for details). In summary, the BNP model with bridging states has the following advantages: (1) The family $\langle \mu_+^I \rangle_{I \in E}$ can be treated in the same way as the usual BNP model, i.e., the model itself has self-similarity (self-consistency) regardless of the size of the observed inputs. (2) If we restrict the space of interest to $\mathcal{X}_I$ (i.e. forget about the auxiliary space $\mathcal{Y}_I$), it becomes equivalent to the exact BNP model on rectangular partitions. More specifically, we have $\mu_X^I(\cdot) = (1/\alpha)\mu_+^I(\cdot \cap \mathcal{X}_I)$.

## 5  Application to relational data analysis

### 5.1  PCRP-based relational model and Bayesian inference

**Relational model -** The PCRP-based relational model is applied to the input observation matrix $\boldsymbol{Z} = (Z_{i,j})_{N \times M}$. We suppose that the input matrix $\boldsymbol{Z}$ consists of categorical elements, i.e., $Z_{i,j} \in \{1, 2, \ldots, D\}$ ($D \in \mathbb{N}$). The generative model can be constructed as follows. First, a random table assignment is performed by a single CRP for all rows and columns of the input matrix (**Step 1** described in Section 3.1). We will denote by $r_i$ ($\in \mathbb{N}$) the index of the table to which the $i$-th row is assigned and by $c_j$ ($\in \mathbb{N}$) the index of the table to which the $j$-th column is assigned. In the process of the table assignment by CRP, whenever a new table is generated, random coordinates for that table on $[0,1] \times [0,1]$ will be drawn from permuton $\gamma$ (**Step 1**). We will denote by $L_t$ ($\in \mathbb{R}^2$) the coordinates assigned to the $t$-th table ($t = 1, 2, \ldots, K$), where $K$ is the number of tables generated by CRP. With the PCRP table assignments $\boldsymbol{T} := (r_1, \ldots, r_N, c_1, \ldots, c_M)$ and the PCRP table coordinates $\boldsymbol{L} := (L_1, \ldots, L_K)$, they can be uniquely transformed into a rectangular partition of the input matrix $\boldsymbol{Z}$ (**Step 2-3**). For simplicity, we denote by $R(\boldsymbol{T}, \boldsymbol{L})$ the rectangular partition derived from $\boldsymbol{T}$ and $\boldsymbol{L}$. Each block indexed by $k$ ($\in \mathbb{N}$) in the rectangular partition $R(\boldsymbol{T}, \boldsymbol{L})$ has a latent Dirichlet random variable $\vartheta_k \sim \text{Dirichlet}(\boldsymbol{\alpha}_0)$ ($k = 1, 2, \ldots, K$), where $\boldsymbol{\alpha}_0 = (\alpha_0, \ldots, \alpha_0)$ is a $D$-dimensional non-negative hyper parameter. Each element $Z_{i,j}$ is drawn from the categorical distribution with the parameter $\vartheta_{\mathbf{k}(i,j)}$, where $\mathbf{k}(i,j)$ indicates the block index to which the entry with the $i$-th row and the $j$-th column belongs. In summary, it can be viewed as a problem of estimating **PCRP table assignments $\boldsymbol{T}$** and **PCRP table coordinates $\boldsymbol{L}$**, given the **input matrix $\boldsymbol{Z}$** and the

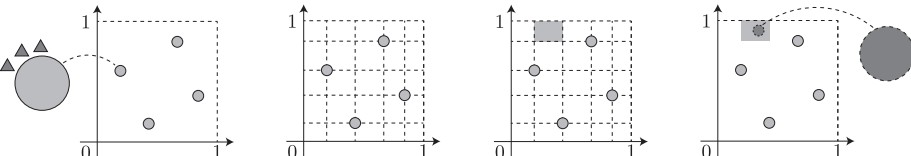

Figure 8: Illustration of generating a new table in Gibbs sampling. **Left**: Coordinates of existing $K$ tables with customers already seated. **Second from left**: $(K+1)^2$ subsections that are candidates for the coordinates from which the new table will be generated. **Third from left**: Subsection (colored in gray) $[v_1, v_2] \times [h_1, h_2]$ chosen as the location where the new table will be generated. **Right**: Coordinates of the new table (colored in dark gray) drawn from permuton $\gamma([v_1, v_2] \times [h_1, h_2])$.

hyper-parameters: **permuton** $\gamma$, **Dirichlet distribution parameters** $\alpha_0 > 0$, and **concentration parameter of PCRP** $\eta > 0$.

**Bayesian inference -** The PCRP relational model can lead to very simple inference algorithms based on the MCMC methods. Following the generative model described above, the joint probability density is expressed as follows:

$$p\left(\boldsymbol{Z}, \boldsymbol{T}, \boldsymbol{L} \mid \gamma, \alpha_0, \eta\right) = p_{\mathrm{CRP}}(\boldsymbol{T} \mid \eta) \cdot \left(\prod_{t=1}^{K} p_{\mathrm{perm.}}(L_t \mid \gamma)\right) \cdot p_{\mathrm{obs.}}(\boldsymbol{Z} \mid R(\boldsymbol{T}, \boldsymbol{L}), \alpha_0), \qquad (1)$$

where the first term $p_{\mathrm{CRP}}(\boldsymbol{T} \mid \eta)$ is the probability distribution for the standard CRP, the second term $p_{\mathrm{perm.}}(L_t \mid \gamma)$ is the probability density that $L_t$ is drawn from the permuton $\gamma$, and the third term is

$$p_{\mathrm{obs.}}(\boldsymbol{Z} \mid R(\boldsymbol{T}, \boldsymbol{L}), \alpha_0) \propto \prod_{k=1}^{K} \left(\frac{\Gamma(D\alpha_0)}{\Gamma(D\alpha_0 + \sum_{d=1}^{D} \mathcal{N}_{k,d})} \prod_{d=1}^{D} \frac{\Gamma(\alpha_0 + \mathcal{N}_{k,d})}{\Gamma(\alpha_0)}\right), \qquad (2)$$

where $\mathcal{N}_{k,d}$ denotes the number of elements in both the $k$-th block and the $d$-th category of the categorical distribution. The overview of the MCMC inference algorithm is to simulate the above joint probability by sequentially updating the table assignments $\boldsymbol{T}$ and the table coordinates $\boldsymbol{L}$.

One simple question is whether PCRP can construct Gibbs sampling in updating table assignments like the usual CRP. Fortunately, PCRP can lead to Gibbs sampling. Updating table assignments for rows and columns involves assigning them to existing tables or generating a new table, similar to the standard CRP. Calculating the probability of the former is straightforward, whereas calculating the latter, i.e., the probability of generating a new table, requires a little more care. Figure 8 shows an illustration of the case where a new table is generated in Gibbs sampling. We suppose that there are currently $K$ tables. The posterior probability of choosing a new table depends on the random coordinates of that table and the random rectangular partition derived from it, which has at most $(K+1)^2$ cases of new rectangular partitions. Specifically, for each of the $K$ tables on $[0,1] \times [0,1]$, we draw a horizontal and vertical crosshair line, respectively, to divide $[0,1] \times [0,1]$ into $(K+1)^2$ subsections (Figure 8, second). The probability of the new table's coordinates falling into each of those $(K+1)^2$ subsections can be calculated using Equation (1). Therefore, the sum of those probabilities can be regarded as the probability that the new table will be generated. This can be calculated exactly without any approximation. Therefore, the conditional posterior distributions for $r_i$ and $c_j$ can be calculated explicitly, and Gibbs sampling can be easily performed. The whole MCMC algorithm will be discussed in the supplementary material. Our code will be available at `https://github.com/nttcslab/permuton-induced-crp`.

## 5.2 Experimental evaluation

We compare PCRP with the existing BNP models for rectangular partitioning, IRM [38], MP [55], RTP [48], and BBP [47]. The detailed experimental setup is described in the supplementary material.

**Datasets -** Four social network datasets [40]: **Wiki** [1], **Facebook** [2], **Twitter** [3], and **Epinions** [4]. All data is public and does not contain any personally identifiable information (See [41] for license). We selected the top 1000 active nodes based on their interactions with others; subsequently we randomly sampled $500 \times 500$ matrix to construct the relational data, as in [25, 47]. We held out 20% cells of the input data for testing, and each model was trained by the MCMC using the

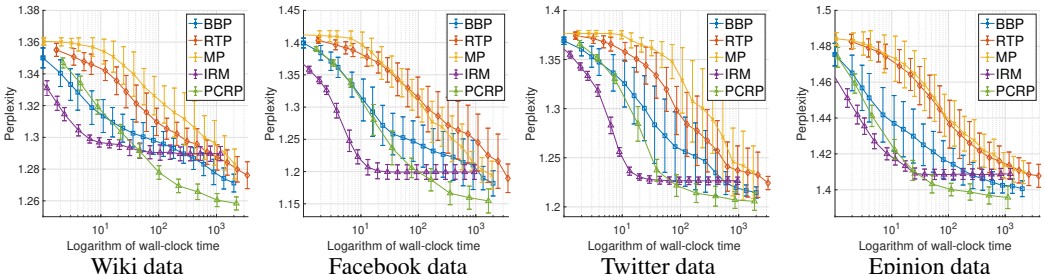

Figure 9: Experimental results of perplexity comparison for four real world data, relationship between test perplexity (mean±std) evolution and logarithm of wall-clock time.

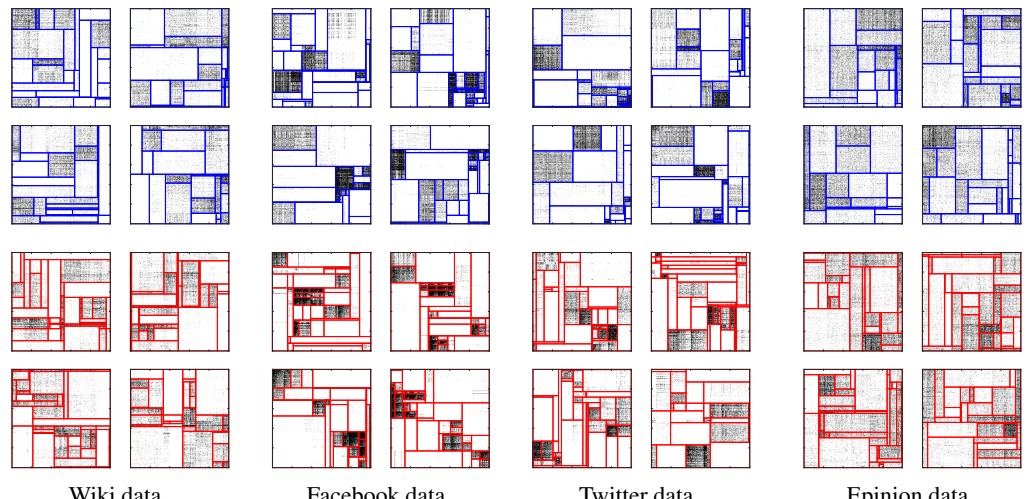

Figure 10: Four examples of analysis results for each data by BBP (**Top**) and PCRP (**Bottom**).

remaining 80% of the cells. We evaluated the models using perplexity as a criterion: $\mathrm{perp}(\hat{Z}) = \exp(-(\log p(\hat{Z}))/E)$, where $E$ is the number of non-missing cells in the partitioned matrix $\hat{Z}$.

**Experimental results -** We ran 10 trials of analysis for each method on each data set. Figure 9 summarizes the test perplexity comparison results. We recall that IRM and MP are limited in the class of rectangular partitions they can represent, and only PCRP, BBP, and RTP can represent arbitrary rectangular partitioning. Indeed, it can be seen that these arbitrary rectangular partitioning models have good prediction accuracy when MCMC converges. Among them, PCRP in particular achieves the best prediction accuracy for all the data. As another perspective, BBP, RTP, and MP are SBP-type models, while PCRP and IRM are CRP-type models. It can be seen that PCRP improves the perplexity from an early stage compared to SBP-like methods. In light of the above, we can confirm that PCRP can reduce the problems of local optima and slow mixing in Bayesian inference. As shown in Figure 10, BBP extracts a very biased rectangular partition due to the fact that the addition of new blocks tends to occur in the lower right corner. In other words, BBP only has a high probability of adding blocks to the lower right, so if a coarse cluster is created in the upper left, it will be difficult to modify the coarse cluster into a fine one. On the other hand, PCRP has the flexibility to add new blocks to the entire area, so it can properly find coarse and fine clusters in the entire area.

## 6 Conclusion and Discussion

**Summary -** This paper has proposed a new stochastic process for relational data analysis. Our main contributions are as follows: (1) We introduce the notion of the permuton to the CRP and obtain a probabilistic model of rectangular partitioning of a matrix. (2) We show a unified framework for PCRP to various classes of rectangular partitioning based on the design of the permuton. (3) We discuss the theoretical imperfection of PCRP for projectivity and exchangeability in terms of rectangular partitioning and propose a new representation of the BNP model as a way to overcome this imperfection that applies PCRP as a bridging state to existing BNP relational models.

## Funding disclosure

Funding in direct support of this work is NTT Corporation, without any third party funding.

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
