# Supplementary Material for "Permuton-induced Chinese Restaurant Process"

**Masahiro Nakano, Yasuhiro Fujiwara, Akisato Kimura, Takeshi Yamada, Naonori Ueda**
NTT Communication Science Laboratories, NTT Corporation
{masahiro.nakano.pr, yasuhiro.fujiwara.kh, akisato.kimura.xn,
takeshi.yamada.bc, naonori.ueda.fr}@hco.ntt.co.jp

## A    Details of permutation classes (Section 2.1)

We will give more details about the *permutation classes* mentioned briefly in the text. To identify a permutation class, we introduce the notion of *occurrences*. We suppose that $\tau$ and $\sigma$ are permutations of size $k$ and $n$, respectively. The *occurrences* of pattern $\tau$ in $\sigma$ is a subsequence $\sigma_{i_1}, \ldots, \sigma_{i_k}$ that is *order-isomorphic* to $\tau$, that is, for all indices $s, t \in [k]$, we have $\sigma_{i_s} < \sigma_{i_t} \iff \tau_s < \tau_t$. We denote $\tau < \sigma$ when $\sigma$ contains $\tau$, that is, $\tau$ is the occurrences of $\sigma$. One way to identify permutation classes is to consider sets of permutations which are closed downward under this containment order. Specifically, $\mathcal{C}$ is a class if for all $\pi$ in $\mathcal{C}$ and all $\sigma \leq \pi$, $\sigma$ is also in $\mathcal{C}$. We can specify permutation classes as closures: if $A$ is any set of permutations, its closure is the permutation class

$$\mathrm{Cl}(A) := \{\sigma : \sigma \leq \pi \text{ for some } \pi \in A\}. \tag{1}$$

However, it is often more popular to specify classes by what they do not contain; for any permutation class $\mathcal{C}$ there is a unique *antichain* $B$ (the details of which are given immediately below) such that

$$\mathrm{Av}(B) := \{\sigma : \sigma \not\geq \pi \text{ for all } \pi \in B\}. \tag{2}$$

For convenience, we introduce useful notation for representing *antichains* as follows. Let $p = p_1 \ldots p_l$ be a permutation and $\hat{p}$ be obtained by inserting a single dash between some adjacent entries of $p$. For example, we have $p = 3142$ and $\hat{p} = 3\text{-}14\text{-}2$. If there exists some subsequence $\sigma(i_1), \ldots, \sigma(i_l)$ of a permutation $\sigma$, then we have the following two properties. First, the relative order of the terms in the subsequence $\sigma(i_1), \ldots, \sigma(i_l)$ matches the relative order of the entries of $\sigma$, that is, for all $j, k \in \{1, \ldots, l\}$, we have that $\sigma(i_j) < \sigma(i_k)$ if and only if $p_j < p_k$. Secondly, if $p_j$ and $p_{j+1}$ are not separated by a dash in $\hat{p}$, then $i_j = i_{j+1} - 1$, that is, $\sigma(i_j)$ and $\sigma(i_{j+1})$ are adjacent in $\sigma$. If $\sigma$ does not contain the pattern $p$, we can regard that it avoids $p$. For example, consider $\sigma = 546312$. The subsequence 5612 is an occurrence of the pattern 3-4-1-2 in $\sigma$, but is not an occurrence of the pattern 3-41-2 since the 6 and 1 are non-adjacent in $\sigma$. In the following, we regard the antichain as a set of patterns $p$ and $\hat{p}$, and introduce some permutation classes with antichains. Specifically, we introduce the following specific classes of permutations:

- **Separable permutation**: $\mathrm{Av}(2413, 3142)$.
- **Separable skew-merged permutation**: $\mathrm{Av}(2143, 2413, 3142, 3412)$.
- **Baxter permutation**: $\mathrm{Av}(2\text{-}41\text{-}3, 3\text{-}14\text{-}2)$.
- **2-clumped permutation**: $\mathrm{Av}(3\text{-}51\text{-}24, 3\text{-}51\text{-}42, 24\text{-}51\text{-}3, 42\text{-}51\text{-}3)$.

### A.1    Separable permutation

The separable permutations are one of the most popular classes of permutations, which has a bijection to hierarchical and diagonal rectangulations. They can be built from the permutation 1 by repeatedly applying two operations, known as direct sum $\oplus$ and skew sum $\ominus$ which are defined, respectively, on

permutations $\pi$ of length $m$ and $\sigma$ of length $n$ by

$$
\pi \oplus \sigma(i) = \begin{cases} \pi(i) & (i = 1, \ldots, m) \\ \sigma(i - m) + m & (i = m + 1, \ldots, m + n) \end{cases}
$$

$$
\pi \ominus \sigma(i) = \begin{cases} \pi(i) + n & (i = 1, \ldots, m) \\ \sigma(i - m) & (i = m + 1, \ldots, m + n) \end{cases}
$$

Separable permutations have a bijection to *hierarchical and diagonal rectangulations* [24, 5], a subset of *hierarchical partitioning*, which are expressed as binary trees where nodes represent a vertical or horizontal separation of a rectangle into two disjoint rectangles.

## A.2   Separable skew-merged permutation

A permutation is said to be *skew-merged* if it is the union of an increasing subsequence and a decreasing subsequence [25, 6]. In this paper, we specifically focus on the class of *separable skew-merged* permutations. Separable skew-merged permutations have a bijection to a special subset of hierarchical partitioning, in particular, those where only one rectangle is allowed to be further cut in each layer. As we will see in the next section, this class of permutations has a very prospective shape in its geometric representation.

## A.3   Baxter permutation

The Baxter permutation has been introduced as a class of permutations in the context of fixed points for the composition of commuting functions [10]. The Baxter permutations have a bijection to *diagonal rectangulations* [26, 5, 14], which has a representative in which every rectangle's interior intersects the diagonal. Recently, the block-breaking process [18] has successfully used this class of permutations to extend the stick-breaking process [23] for sequence partitioning to the stochastic process for rectangular partitioning.

## A.4   $2$-clumped permutation

This class of permutations has received particular attention in recent years due to its close relationship with rectangular partitioning. This class of permutations has the following properties [20]. A pair $\sigma_i$ and $\sigma_{i+1}$ of a permutation $\sigma$ is a *descent* of $\sigma$ if $\sigma_i > \sigma_{i+1}$. For every descent of $\sigma$, a *clump* is defined as a maximal set of consecutive values $a, a + 1, \ldots, b$ with $\sigma_i > b > a > \sigma_{i+1}$ such that either all elements of $\{a, a + 1, \ldots, b\}$ occur to the left of the descent or all elements of $\{a, a + 1, \ldots, b\}$ occur to the right of the descent. For example, consider a permutation $167439285$. The pair $92$ is just a descent of the permutation $167439285$. Four clumps are associated with this descent, $\{3, 4\}$, $\{5\}$, $\{6, 7\}$, and $\{8\}$. A permutation $\sigma$ is a $k$-clumped permutation if every descent of $\sigma$ has at most $k$ associated clumps. The permutation $167439285$ is $k$-clumped for any $k \geq 4$ because four clumps are associated with the descent $92$ and fewer clumps are associated with any other descent of the permutation. The 2-clumped permutation corresponds to the case $k = 2$. Interestingly, the 2-clumped permutations have a bijection to *generic rectangulations* [20].

## B   Details of permuton designs for PCRP (Section 3.2)

In this section, we discuss the method of designing permuton for PCRP. First of all, in practical terms, we recommend employing the uniform permuton $\gamma(\cdot) = \mathrm{Leb}(\{\cdot\})$ because it is concise and useful. More specifically, the advantages of PCRP with the uniform permuton can be listed as follows:

- PCRP with the uniform permuton can represent arbitrary rectangular partitioning, i.e., generic rectangulations.
- The inference algorithm can be easily derived.

On the other hand, the disadvantage of PCRP with the uniform permuton is that the probabilities to all generic rectangulations are not uniform and has some bias, as mentioned in the body of the paper. Therefore, from a theoretical point of view, in order to compensate for this disadvantage, we need to design a permuton that is uniformly distributed for some class of rectangular partitioning.

We start with the relationship between permutation classes, rectangular partitioning classes, and the existing BNP model. Figure 1 illustrates the overall picture:

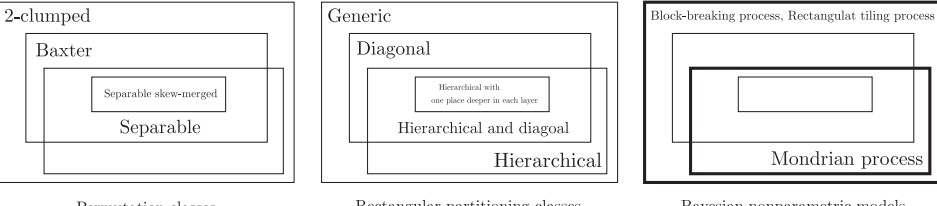

Figure 1: Relationship between permutation classes, rectangular partitioning classes and Bayesian nonparametric models.

The inclusion relations in the set of figures represent the inclusion relations of the respective classes. As shown by the existing BNP models such as BBP [18], RTP [19], and MP [22, 21], the most important rectangular partitioning classes in terms of applications in relational data analysis are hierarchical partitionings and generic rectangulations. Unfortunately, to the best of our knowledge, however, the permutation class with the one-to-one correspondence with hierarchical partitioning are still under development, and we do not know how to handle uniform hierarchical partitioning explicitly through permuton. As for generic rectangulations, the permutation class with one-to-one correspondence has been actively studied as 2-clumped permutations. However, to the best of our knowledge, there is still no way to construct the corresponding permuton to uniform 2-clumped permutations. This means that it is currently tricky to explicitly design a clever permuton that will always restrict its samples to 2-clumped permutations. Although these facts are certainly regrettable, the research on permutons is now undergoing rapid development, and it is strongly expected that methods for handling hierarchical partitionings and generic rectangulations through permutons will be developed in the near future. In the remainder of this section, we introduce several permutation classes for which permuton construction methods are now explicitly known. And finally, we discuss how permutons can be constructed for generic rectangulations from a practical point of view.

### B.1  Explicit permuton representation for some permutation classes

**Hierarchical partitioning with one place deeper in each layer -** Uniform partitioning of this class can be obtained by restricting the random permutation of coordinates in the CRP tables to uniform separable skew-merged. Figure 2 shows illustrations of geometric representations of the separable skew-merged permutations. If $\sigma_n$ is a uniform separable skew-merged permutation of size $n$, then $\gamma_{\sigma_n}$ converges in distribution to a deterministic permuton $\gamma$ (See, e.g., Theorem 3.3 and Section 3.2.1 [9]): For all Borel measurable set $A$ of $[0,1] \times [0,1]$, We have

$$\gamma(\cdot) = \frac{1}{\sqrt{2}}\text{Leb}\Big(\big\{(v,w) \in \cdot : v + w = 0\big\}\Big) + \frac{1}{\sqrt{2}}\text{Leb}\Big(\big\{(v,w) \in \cdot : v - w = 0\big\}\Big). \tag{3}$$

Intuitively, the permuton $\gamma$ has an X-shaped density on $[0,1] \times [0,1]$.

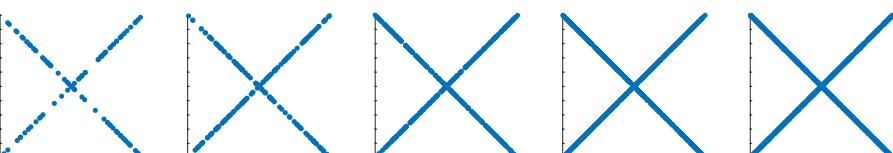

Figure 2: Examples of geometric representations of separable skew-merged permutation $\sigma$. From left to right, $|\sigma| = 100$, $|\sigma| = 200$, $|\sigma| = 400$, $|\sigma| = 800$, and $|\sigma| = 1600$.

**Hierarchical and diagonal rectangulation -** Uniform hierarchical and diagonal rectangulations can be expressed by uniform separable permutations. If $\sigma_n$ is a uniform separable permutation of size $n$, then $\gamma_{\sigma_n}$ converges in distribution to the *Brownian separable permuton* [8, 7, 16]. We consider a Brownian path $(B_t;\ t \geq 0)$. Then, *normalized Brownian excursion* $(e_t;\ 0 \leq t \leq 1)$ is defined by

$$e_t := \frac{1}{\sqrt{d_1 - g_1}} \left| B_{g_1 + t(d_1 - g_1)} \right|, \tag{4}$$

where $g_1 = \sup\{t < 1; \ B_t = 0\}$ and $d_1 = \inf\{t > 1; \ B_t = 0\}$. We additionally introduce a function $F$ assigning balanced independent signs $\{+, -\}$ on the local minima of $e_t$, and call $(e_t, F)$ the *signed Brownian excursion*. See also Figure 3.

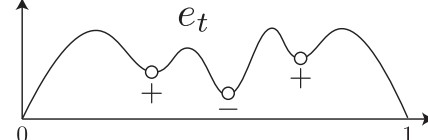

Figure 3: Illustration of signed Brownian excursion $(e_t, F)$.

Finally, using the Lebesgue preserving function $\varphi : [0, 1] \to [0, 1]$ such that $(a, b) \subset [0, 1]$ is an inversion if and only if the sign of $\min_{t \in [a,b]} e(t)$ (i.e., $F(\min_{t \in [a,b]} e(t))$) is $-$, we obtain the Brownian separable permuton:

$$\gamma(\cdot) = \mathrm{Leb}\big\{t \in [0, 1]; (t, \varphi(t)) \in \cdot\big\}. \tag{5}$$

In addition, we would like to emphasize that one of the most essential properties of the Brownian separable permuton is self-similarity (See Theorem 1.6 [16]).

**Diagonal rectangulation -** Uniform diagonal rectangulations can be expressed by uniform Baxter permutations. Figure 4 shows illustrations of geometric representations of the Baxter permutations. If $\sigma_n$ is a uniform Baxter permutation of size $n$, then $\gamma_{\sigma_n}$ converges in distribution to the *Baxter permuton* [12]. The construction of the Baxter permuton is quite complicated, involving bipolar orientations and walks in the quadrant, therefore we only sketch it here. Let $e_t = (e_{t,1}, e_{t,2})$ be a 2-dimensional Brownian excursion of correlation $(-1/2)$ conditioned to stay in the non-negative quadrant. We introduce a family of stochastic differential equations indexed by $u \in \mathbb{R}$:

$$\begin{cases} dZ^{(u)}(t) = \mathbb{I}\big[Z^{(u)}(t) > 0\big]de_{t,2} - \mathbb{I}\big[Z^{(u)}(t) \geq 0\big]de_{t,1} & (t \leq u) \\ Z^{(u)}(t) = 0 & (t \geq u) \end{cases} \tag{6}$$

and obtain solutions $Z_e$. Then, we build the following relation $\leq_Z : i \leq_Z j \iff \{i < j \text{ and } Z^{(i)}(j) < 0\}$ or $\{i > j \text{ and } Z^{(i)}(j) \geq 0\}$ or $\{i = j\}$. Finally, we obtain the Baxter permuton:

$$\gamma(\cdot) = \mathrm{Leb}\big\{t \in [0, 1]; (t, \phi_{Z_e}(t)) \in \cdot\big\}, \tag{7}$$

where $\phi_{Z_e}(t) = \{u \in [0, 1]; u \leq_{Z_e} \leq t\}$.

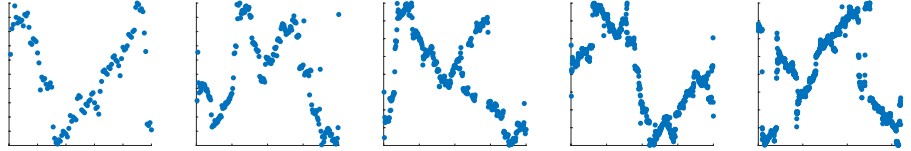

Figure 4: Examples of geometric representations of Baxter permutation $\sigma$. From left to right, $|\sigma| = 100$, $|\sigma| = 200$, $|\sigma| = 400$, $|\sigma| = 800$, and $|\sigma| = 1600$.

### B.2 Indirect permuton representation for 2-clumped permutations

As described above, the construction of permutons for various permutation classes is still in the process of development, and more permutons for various permutation classes are sure to be discovered in the future. On the other hand, from the viewpoint of the practical use of relational data analysis, we currently need some way to handle uniform 2-clumped permutations corresponding to generic rectangulations. Motivated by this, we would like to discuss two possibilities.

**(1) Enumeration of generic rectangulations -** One way to obtain a uniform random 2-clumped permutation is to use a enumeration algorithm that has been discovered very recently [17]. This enumeration algorithm directly use generic rectangulations, according to the one-to-one correspondence between 2-clumped permutations and generic rectangulations, and use the idea of *insertion*. See Figure 6. The idea of insertion is to add a new rectangle into the bottom-right corner of the rectangulation (Figure 6, left). Given the rectangulation with $n$ blocks, we first define a set of points that can become the top-left corner of the newly added rectangle. If any rectangle $r$ of the given

rectangulation is tangent to the lower boundary of the outer rectangle, we consider all edges that form the left side of $r$ and select one interior point (circles in Figure 6, second from left) from such edges. Similarly, for any rectangle $r$ tangent to the right boundary of the outer rectangle, we consider the set of all edges forming the top edge of $r$ and select one interior point (circles in Figure 6, second from left) from such edges. Then we can choose one point $v$ uniformly in the set of such interior points. If $v$ is a vertical insertion point, then the new rectangulation is obtained from the current rectangulation by inserting a new rectangle in the lower right corner. In this case, the new rectangle will be located to the right of $v$, with all rectangles tangent to the bottom boundary of the outer rectangle exactly above it, and the new rectangle have all rectangles tangent to the vertical wall through $v$ below $v$ exactly to the left. Similarly, if $v$ is a horizontal insertion point, the new rectangulation can be obtained from the current rectangulation by inserting a new rectangle in the lower right corner. Then the new rectangle have all the rectangles that are below $v$ and tangent to the right boundary of the outer rectangle exactly to the left, and the new rectangle will have all the rectangles in the outer rectangle that are to the right of $v$ and tangent to the horizontal wall through $v$ exactly to the top (Figure 6, right).

**(2) Use of rejection sampling -** Another way is to use rejection sampling to replace permuton, where the descriptive method is not known but the probability ratio of each 2-clumped permutation sample is known. Specifically, using $\gamma'(\cdot) = \mathrm{Leb}(\{\cdot\})$ as the proposal distribution instead of the true permuton $\gamma$, if the generated permutation does not result in a 2-clumped permutation, the sample is rejected and the process is repeated until a 2-clumped permutation is generated. The advantage of this method is that it is very easy to implement. On the other hand, the disadvantage is that as the length of the permutation increases, the number of times it is rejected must increase, as Figure 5 implies. Therefore, it is practical to use enumeration algorithms together when the size of the permutation becomes large.

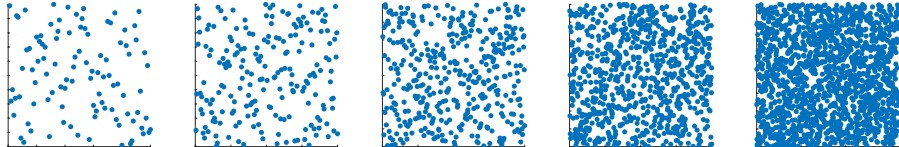

Figure 5: Examples of geometric representations of uniform permutation $\sigma$. From left to right, $|\sigma| = 100$, $|\sigma| = 200$, $|\sigma| = 400$, $|\sigma| = 800$, and $|\sigma| = 1600$.

## C Details of projectivity and exchangeability for PCRP (Section 4)

This section provides a detailed explanation for Section 4, **Intermediate level representation between SBP and CRP**, in the main body of our paper. To emphasize again, just to be clear, the discussion in this section is not primarily motivated by practical issues in relational data analysis, but rather to be consistent with the theory of Bayesian nonparametrics. PCRP has become a probabilistic model with two aspects: CRP table assignment and rectangular partitioning. As a model of CRP table assignment, PCRP has projectivity and exchangeability, but as a rectangular partitioning model, it loses projectivity and exchangeability. Therefore, we propose a strategy on how to restore the projectivity and exchangeability to PCRP as a rectangular partitioning model. First, we introduce a precise description of *rectangular partitioning* of matrices indicated by the index set and the projectors that connect large matrices to small ones. Second, we identify two concepts that are important properties of the BNP model, namely, *exchangeability* and *projectivity*. Thirdly, we point out that PCRP does not preserve these conditions in terms of rectangular partitioning of matrices. Finally, to solve this problem, we apply PCRP to the bridging condition of the exact BNP model of rectangular partitioning.

*Index set* - Consider a set of rectangular partitioning of a matrix whose rows and columns are indexed by natural numbers. As the index set $E$, we will deal with the set of matrices $I = I^{(r)} \times I^{(c)}$ ($I \in E$), where $I^{(r)}$ and $I^{(c)}$ are sets of natural numbers, which correspond to the indices of the rows and columns, respectively. We here emphasize that the orders of the rows and columns are not fixed. Consider two pairs $I = I^{(r)} \times I^{(c)}$ and $J = J^{(r)} \times J^{(c)}$. We define the partial order $I \preceq J$ as an inclusion, that is, $I^{(r)} \subseteq J^{(r)}$ and $I^{(c)} \subseteq J^{(c)}$. For example, $I^{(c)} = \{1, 2, 6, 7\} \subset J^{(c)} = \{1, 2, 5, 6, 7\}$. We write $(v, h)$-cell to denote an element of $I$ whose row and column are indexed by $v \in I^{(r)}$ and $h \in I^{(c)}$, respectively.

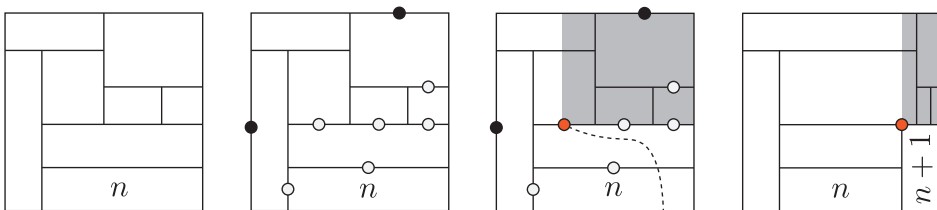

Figure 6: Illustration of enumeration of 2-clumped permutations via generic rectangulations.

*Rectangular partitioning* - Plainly, rectangular partitioning can be regarded as making a set of cluster assignments of all cells of $I$ such that, given suitable orders of rows $I^{(r)}$ and columns $I^{(c)}$, all clusters form rectangle blocks. Let $\mathcal{X}_I$ be the collection of all rectangular partitions of $I$. Each element $x_I \in \mathcal{X}_I$ can be expressed as an equivalence relation $x_I : I^{(r)} \times I^{(c)} \times I^{(r)} \times I^{(c)} \to \{0, 1\}$ such that $x_I(i, j, i', j') = 1$ if and only if the $(i, j)$-cell and the $(i', j')$-cell are in the same cluster.

*Projector* - The mapping $Q_{J,I} : \mathcal{X}_J \to \mathcal{X}_I$ restricts a sample $x_J \in \mathcal{X}_J$ of rectangular partitioning of $J$ by keeping the $I$ entries unchanged and removing the remaining entries. The projection of a measure $\mu_I$ on $(\mathcal{X}_I, \mathbf{2}^{\mathcal{X}_I})$ is also defined by means of a push-forward, $(Q_{J,I}\mu_J)(A_I) := \mu_J(Q_{J,I}^{-1}A_I)$, for any $A_I \in \mathbf{2}^{\mathcal{X}_I}$.

Now that we have defined a space of rectangular partitions whose dimension changes according to the size of the input matrix, we will reveal a BNP model that can handle them in a unified manner. We consider probability measures $\mu_I$ on $(\mathcal{X}_I, \mathbf{2}^{\mathcal{X}_I})$, and hope to find the family $\langle \mu_I, Q_{J,I} \rangle_{I \prec J \in E}$ which satisfy the following conditions.

(C1) **Exchangeability:** For any $I = I^{(r)} \times I^{(c)} \in E$, and any permutation $\sigma : I^{(r)} \to I^{(r)}$ and $\sigma' : I^{(c)} \to I^{(c)}$, we consider $I' = \sigma(I^{(r)}) \times \sigma'(I^{(c)}) \in E$. Then, for any $y_{I^{(r)} \times I^{(c)}} \in \mathcal{Y}_I$, we have $\mu_X^I(x_{I^{(r)} \times I^{(c)}} \in \mathcal{X}_X^I) = \mu_X^{I'}(x_{\sigma(I^{(r)}) \times \sigma(I^{(c)})} \in \mathcal{X}_X^{I'})$.

(C2) **Projectivity:** For any pair $I \prec J \in E$, we have $\mu_X^I(x_I) = \mu_X^J((Q_X^{J,I})^{-1}x_I)$.

The model that satisfies the two conditions mentioned above is called an infinitely exchangeable model, and is generally applied to machine learning as a BNP relational model. In fact, IRM [15], the MP-based model [22, 21], the RTP-based model [19], and the BBP-based model [18], etc. all satisfy these conditions. Our first concern here is whether PCRP really satisfies these conditions or not. Unfortunately, unless we choose a special permuton, PCRP does not satisfy these conditions:

**Remark C.1** *PCRP using the uniform permuton, the Brownian separable permuton, and the Baxter permuton does not satisfy the conditions of exchangeability (C1) and projectivity (C2). The reason for this is that the addition of a new table in the CRP table assignment perspective modifies and affects previous partitions in the rectangular partitioning perspective.*

Therefore, in this section, PCRP is treated not as a probabilistic model on rectangular partitioning of a matrix $I$, but as a probabilistic model on table assignment of rows $I^{(r)}$ and columns $I^{(c)}$ of $I$. As the auxiliary space $\mathcal{Y}_I$, we introduce the table assignments of $I^{(r)}$ and $I^{(c)}$ by CRP and a set of coordinates on $[0, 1] \times [0, 1]$ of the tables. The corresponding probability model $\mu_Y^I$ on $(\mathcal{Y}_I, \mathbf{2}^{\mathcal{Y}_I})$ should be PCRP. The projector $Q_Y^{J,I} : \mathcal{Y}_J \to \mathcal{Y}_I$ restricts a sample $y_J \in \mathcal{Y}_J$ (i.e., a sample of table assignments of $J^{(r)}$ and $J^{(c)}$) by keeping the $I^{(r)}$ and $I^{(c)}$ entries unchanged removing the remaining entries, and deleting tables where no one is seated. Then, connecting the bridging state in the auxiliary space $\mathcal{Y}_I$ with samples in $\mathcal{X}_I$, we consider the combined probability model over the union space $\mathcal{X}_I \cup \mathcal{Y}_I$. To address probabilistic models on the union space $\mathcal{X}_I \cup \mathcal{Y}_I$, we modify the previous probabilistic models $\mu_X^I$ and $\mu_Y^I$ as follows:

$$\hat{\mu}_X^I(\cdot \in \mathbf{2}^{\mathcal{X}_I \cup \mathcal{Y}_I}) := \mu_X^I(\cdot \cap \mathcal{X}_I) \quad \text{and} \quad \hat{\mu}_Y^I(\cdot \in \mathbf{2}^{\mathcal{X}_I \cup \mathcal{Y}_I}) := \mu_Y^I(\cdot \cap \mathcal{Y}_I). \tag{8}$$

For simplicity of notation, the modified probabilistic models $\hat{\mu}_X^I$ and $\hat{\mu}_Y^I$ will be denoted as original $\mu_X^I$ and $\mu_Y^I$ in the following. Finally, we consider a probabilistic model $\mu_+^I$ in form of

$$\mu_+^I = \alpha \mu_X^I + (1 - \alpha)\mu_Y^I, \tag{9}$$

where $0 \leq \alpha \leq 1$ is a tunable real variable. Fortunately, we can treat this new model $\mu_+^I$ as a BNP model, that is, there uniquely exists a projective limit of $\mu_+^I$ ($I \to \mathbb{N} \times \mathbb{N}$):

**Theorem C.2** *Two families $\langle \mu_X^I, Q_X^{J,I} \rangle_{I \prec J \in E}$ and $\langle \mu_Y^I, Q_Y^{J,I} \rangle_{I \prec J \in E}$ of BNP models are described as above. Then, there uniquely exists projective limit probability measures $\mu_X^E$ and $\mu_Y^E$ of $\langle \mu_X^I, Q_X^{J,I} \rangle_{I \prec J \in E}$ and $\langle \mu_Y^I, Q_Y^{J,I} \rangle_{I \prec J \in E}$, respectively. Moreover, we consider the combined model give by $\mu_+^I = \alpha \mu_X^I + (1 - \alpha)\mu_Y^I$, and construct a projective system $\langle \mu_+^I, Q_+^{J,I} \rangle_{I \prec J \in E}$, where $Q_+^{J,I} : \mathcal{X}_I \cup \mathcal{Y}_I \to \mathcal{X}_I \cup \mathcal{Y}_I$ is defined as*

$$Q_+^{J,I}(A_J) := Q_X^{J,I}(A_J \cap \mathcal{X}_I) \cup Q_Y^{J,I}(A_J \cap \mathcal{Y}_I), \tag{10}$$

*for any $A_J \in 2^{\mathcal{X}_I \cup \mathcal{Y}_I}$. Then, there uniquely exists the projective limit probability measure $\mu_+^E$ of the projective system $\langle \mu_+^I, Q_+^{J,I} \rangle_{I \prec J \in E}$ (The projector of the measures is also defined employing a push-forward) as:*

$$\mu_+^E = \alpha \mu_X^E + (1 - \alpha)\mu_Y^E. \tag{11}$$

*Furthermore, for any $I = I^{(r)} \times I^{(c)} \in E$ and any permutation $\sigma : I^{(r)} \to I^{(r)}$ and $\sigma' : I^{(c)} \to I^{(c)}$, the probability measure $\mu_+^I$ satisfies the exchangeability condition (C1).*

**Proof** We show that (a) there uniquely exists the projective limits $\mu_X^E$ and $\mu_Y^E$ of $\langle \mu_X^I, Q_X^{J,I} \rangle_{I \prec J \in E}$ and $\langle \mu_Y^I, Q_Y^{J,I} \rangle_{I \prec J \in E}$, respectively, (b) there uniquely exists the projective limit $\mu_+^E$ of $\langle \mu_+^I, Q_+^{J,I} \rangle_{I \prec J \in E}$, and (c) the probability measure $\mu_+^I$ is exchangeable. As a sketch of the proof, the former two issues can be obtained by applying Kolmogorov's extension theorem [11] to the projective system of each model. The last issue is immediately obtained from exchangeability of $\mu_X^I$ and $\mu_Y^I$.

*(a) Unique existence of projective limits $\mu_X^E$ and $\mu_Y^E$ -* By definition, the projective system $\langle \mu_X^I, Q_X^{J,I} \rangle_{I \prec J \in E}$ has the projectivity condition (C1). More specifically, for any $x_I \in \mathcal{X}_I$ we have

$$\mu_X^I(x_I) = \mu_X^J((Q_X^{J,I})^{-1}x_I). \tag{12}$$

Thus, owing to Kolmogorov's extension theorem, there uniquely exists the projective limit $\mu_X^E$. For the projective system $\langle \mu_Y^I, Q_Y^{J,I} \rangle_{I \prec J \in E}$, we can ignore the random coordinates of CRP tables, and then reduce PCRP to the standard CRP. Owing to the projectivity condition for CRP, we can obtain

$$\mu_Y^I(y_I) = \mu_Y^J((Q_Y^{J,I})^{-1}y_I), \tag{13}$$

for any $y_I \in \mathcal{Y}_I$. As a result, we apply Kolmogorov's extension theorem to $\langle \mu_Y^I, Q_Y^{J,I} \rangle_{I \prec J \in E}$, and obtain the unique projective limit $\mu_Y^E$.

*(b) Unique existence of projective limit $\mu_+^E$ -* Owing to Equations (12), (13), and (10), for any $z_I \in \mathcal{X}_I \cup \mathcal{Y}_I$, we have

$$\begin{aligned} \mu_+^I(z_I) &= \alpha \mu_X^I(z_I) + (1 - \alpha)\mu_Y^I(z_I) \\ &= \alpha \mu_X^I(z_I \cap \mathcal{X}_I) + (1 - \alpha)\mu_Y^I(z_I \cap \mathcal{Y}_I) \\ &= \alpha \mu_X^J((Q_X^{J,I})^{-1}(z_I \cap \mathcal{X}_I)) + (1 - \alpha)\mu_Y^J((Q_Y^{J,I})^{-1}(z_I \cap \mathcal{Y}_I)) \\ &= \mu_+^J((Q_+^{J,I})^{-1}z_I). \end{aligned} \tag{14}$$

Therefore, we apply Kolmogorov's extension theorem to the projective system $\langle \mu_+^I, Q_+^{J,I} \rangle_{I \prec J \in E}$, and obtain the unique projective limit $\mu_+^E$.

*(c) Exchangeability of $\mu_+^I$ -* By construction, the probability measure $\mu_X^I$ is exchangeable, according to the AHK representation. Moreover, similar to the standard CRP, the probability measure $\mu_Y^I$ is also exchangeable. As a result, we can see that $\mu_+^I$ is exchangeable. ∎

# D  Bayesian inference for PCRP-based relational model (Section 5)

In the main text, we have shown a Bayesian inference method for the PCRP-based relational model using Gibbs sampling. As another method that is very easy to implement, we introduce here an

inference method using the Metropolis–Hastings algorithm. The overall picture of the PCRP-based relational model presented in the text is posted again in order to clarify the notations.

**Relational model -** The PCRP-based relational model is applied to the input observation matrix $\boldsymbol{Z} = (Z_{i,j})_{N \times M}$. We suppose that the input matrix $\boldsymbol{Z}$ consists of categorical elements, i.e., $Z_{i,j} \in \{1, 2, \ldots, D\}$ ($D \in \mathbb{N}$). The generative model can be constructed as follows. First, a random table assignment is performed by a single CRP for all rows and columns of the input matrix. We will denote by $r_i$ ($\in \mathbb{N}$) the index of the table to which the $i$-th row is assigned and by $c_j$ ($\in \mathbb{N}$) the index of the table to which the $j$-th column is assigned. In the process of the table assignment by CRP, whenever a new table is generated, random coordinates for that table on $[0, 1] \times [0, 1]$ will be drawn from permuton $\gamma$. We will denote by $L_t$ ($\in \mathbb{R}^2$) the coordinates assigned to the $t$-th table ($t = 1, 2, \ldots, K$), where $K$ is the number of tables generated by CRP. With the PCRP table assignments $\boldsymbol{T} := (r_1, \ldots, r_N, c_1, \ldots, c_M)$ and the PCRP table coordinates $\boldsymbol{L} := (L_1, \ldots, L_K)$, they can be uniquely transformed into a rectangular partition of the input matrix $\boldsymbol{Z}$. For simplicity, we denote by $R(\boldsymbol{T}, \boldsymbol{L})$ the rectangular partition derived from $\boldsymbol{T}$ and $\boldsymbol{L}$. Each block indexed by $k$ ($\in \mathbb{N}$) in the rectangular partition $R(\boldsymbol{T}, \boldsymbol{L})$ has a latent Dirichlet random variable $\vartheta_k \sim \text{Dirichlet}(\boldsymbol{\alpha}_0)$ ($k = 1, 2, \ldots, K$), where $\boldsymbol{\alpha}_0 = (\alpha_0, \ldots, \alpha_0)$ is a $D$-dimensional non-negative hyper parameter. Each element $Z_{i,j}$ is drawn from the categorical distribution with the parameter $\vartheta_{\mathbf{k}(i,j)}$, where $\mathbf{k}(i, j)$ indicates the block index to which the entry with the $i$-th row and the $j$-th column belongs. In summary, it can be viewed as a problem of estimating

- **PCRP table assignments -** $\boldsymbol{T} = (r_1, \ldots, r_N, c_1, \ldots, c_M)$.
- **PCRP table coordinates -** $\boldsymbol{L} = (L_1, \ldots, L_K)$.

given the input data and the hyper-parameters:

- **Input matrix -** $\boldsymbol{Z} := \{Z_{i,j} \mid i = 1, \ldots, N, \ j = 1, 2, \ldots, M\}$, consisting of categorical elements, $Z_{i,j} \in \{1, 2, \ldots, D\}$ ($D \in \mathbb{N}$).
- **Permuton -** $\gamma$, the probability measure on $[0, 1] \times [0, 1]$.
- **Dirichlet distribution parameters -** $\alpha_0 > 0$.
- **Concentration parameter of PCRP -** $\eta > 0$.

Then, the joint probability density is expressed as follows:

$$p(\boldsymbol{Z}, \boldsymbol{T}, \boldsymbol{L} \mid \gamma, \alpha_0, \eta) = p_{\text{CRP}}(\boldsymbol{T} \mid \eta) \cdot \left( \prod_{t=1}^{K} p_{\text{perm.}}(L_t \mid \gamma) \right) \cdot p_{\text{obs.}}(\boldsymbol{Z} \mid R(\boldsymbol{T}, \boldsymbol{L}), \alpha_0), \quad (15)$$

where the first term $p_{\text{CRP}}(\boldsymbol{T} \mid \eta)$ is the probability distribution for the standard CRP, the second term $p_{\text{perm.}}(L_t \mid \gamma)$ is the probability density that $L_t$ is drawn from the permuton $\gamma$, and the third term is

$$p_{\text{obs.}}(\boldsymbol{Z} \mid R(\boldsymbol{T}, \boldsymbol{L}), \alpha_0) \propto \prod_{k=1}^{K} \left( \frac{\Gamma(D\alpha_0)}{\Gamma(D\alpha_0 + \sum_{d=1}^{D} \mathcal{N}_{k,d})} \prod_{d=1}^{D} \frac{\Gamma(\alpha_0 + \mathcal{N}_{k,d})}{\Gamma(\alpha_0)} \right), \quad (16)$$

where $\mathcal{N}_{k,d}$ denotes the number of elements in both the $k$-th block and the $d$-th category of the categorical distribution.

One way to perform the Bayesian inference algorithms is to repeat the following two update rules:

- **Update of table coordinates -** For every table where one or more customers are seated, $t = 1, 2, \ldots, K$, we generate new candidate table coordinates $L_t$ from the permuton $\gamma$ (whose corresponding generic rectangulation also needs to be updated, as discussed in Section 3 in the main text), and decide whether to accept or reject it by the Metropolis-Hastings (MH) algorithm using Equation (15).
- **Update of table assignments -** For every row $i = 1, \ldots, N$ and column $j = 1, \ldots, M$ of the input matrix, we generate a new candidate table assignment $r_i$ and $c_j$ for itself from the standard CRP prior except itself, and decide whether to accept or reject it by the MH algorithm using Equation (15). If a new table is generated, then its random coordinate on $[0, 1] \times [0, 1]$ is drawn from the permuton $\gamma$ (whose corresponding generic rectangulation also needs to be updated, as discussed in Section 3 in the main text).

It is important to note that whenever the table coordinates of PCRP are updated, the corresponding rectangular partitioning itself is also updated based on the PCRP generative model in Section 3 of the main text.

# E    Details of the experimental setup (Section 5)

**Datasets -** We used four social network datasets [27]: (1) **Wiki** (top-left) [1], consisting of 7115 nodes and 103689 edges with diameter 7. (2) **Facebook** (top-right) [2], consisting of 4039 nodes and 88234 edges with diameter 8. (3) **Twitter** (bottom-left) [3], consisting of 81306 nodes and 1768149 edges with diameter 7. (4) **Epinion** (bottom-right) [4], consisting of 75879 nodes and 508837 edges with diameter 14. For each data, we selected the top 1000 active nodes based on their interactions with others; subsequently we randomly sampled $500 \times 500$ matrix to construct the relational data, as in [13]. For model comparison, we held out $20\%$ cells of the input data for testing, and each model was trained by the MCMC using the remaining $80\%$ of the cells.

**Relational models -** We compare the PCRP-based relational model with the BNP stochastic block models based on rectangular partitioning: (1) **IRM** [15]: We employ the product of CRPs, whose concentration parameters are drawn from the $\mathrm{Gamma}(1, 1)$ prior, as in [13, 18]. (2) **MP** [22]: the intermediate random function of the AHK representation is drawn from the MP, the budget parameter of which is set to 3, as in [13, 18]. (3) **RTP** [19]: we combine the product of SBPs (also used in the aforementioned IRM) and the RTP is combined to construct the AHK representation, as in [18]. For the concentration parameters of SBPs are drawn from the $\mathrm{Gamma}(1, 1)$ prior. (4) **BBP** [18]: For all parameters, we used the default settings provided by the original code [18].

## Ethical perspective

Our work does not encourage unethical aspects of machine learning technologies. We are genuinely pursuing the development of Bayesian methods in a number of application settings. However, as is often the case with clustering methods, our proposal can be misused in a variety of situations. Since the PCRP-based relational model may reveal hidden clusters from any input matrices, unanticipated cues can lead to unanticipated results. This issue highly depends on the choice of input data. Therefore, what is suitable as input data needs to be carefully considered from an ethical perspective.