# OpenReview forum: "Permuton-induced Chinese Restaurant Process"
_NeurIPS.cc/2021/Conference — NeurIPS 2021 Poster_

### Official Review · Reviewer_5pQM · 2021-07-13

**Rating:** 7
**Confidence:** 3

**Summary:**

This paper proposed a model for relational data -- permuton-induced Chinese restaurant process, which can be regarded as a multidimensional Chinese restaurant process. The permuton is used to fill in the input matrix according to a Chinese restaurant scheme. The underlying model is well designed for rectangular partitioning, and various empirical results are also provided.

**Limitations And Societal Impact:**

The authors have adequately addressed the limitations and potential negative societal impact of their work.

**Main Review:**

This paper proposed a new model to deal with relational data -- permuton-induced Chinese restaurant process. This model combines the idea of permuton with Chinese restaurant process for different hierarchies. In particular, the permuton-induced Chinese restaurant process is well designed for the problem of rectangular partitioning. A detailed description of the model, as well as how to choose the "hyperparameter"--permuton is explained in detail. The topic of the paper is important. The idea of the paper is novel, and of high quality. The paper is also well-written. I pretty like the idea and presentation of the paper, and would regard this as a good contribution to Neurips community.

**Time Spent Reviewing:**

1 hour

---

> ### Author Response · Authors · 2021-08-10
> **Response to Reviewer 5pQM**
>
> We would like to thank you again for your careful reading of our paper and very positive feedback. In particular, we are happy that you liked the idea of introducing permuton to model Bayesian nonparametric rectangular partitioning. You have evaluated our paper in each of the following aspects:
> - For novelty and quality, you recognized that our model and ideas were novel and of high quality. ​
> - For significance, you found that the topic of Bayesian nonparametric relational data analysis is important, and our work will make a good contribution to the NeurIPS community
> - For clarity, you liked our presentation, and recognized that our paper is also well-written.
>
> We are very encouraged by these feedbacks. This year's NeurIPS is a new attempt to respond during the reviewers' discussions, so if you have any concerns, please let us know, and we will be pleased to respond.

---

### Official Review · Reviewer_rRnG · 2021-07-15

**Rating:** 6
**Confidence:** 4

**Summary:**

The paper introduces the permuton-induced Chinese restaurante process (PCRP), a prior over rectangular partitions of 2D space, and applies the prior to analyze relational data. The sample paths of the prior (Figure 2) exhibit different behavior depending on the permuton measure gamma, which is one of the design choices in the prior. In numerical experiments, probabilistic models using the proposed prior perform competitively with existing choices of prior in terms of held-out perplexity.

**Limitations And Societal Impact:**

For future work, a careful synthetic example to highlight the benefits of PCRP over BBP would be useful. Based on Figure 1c), the types of partition in sample paths of the BBP is more restricted then the types of partitions in sample paths of the PCRP Figure 2. If the BBP is used to analyze data from a PCRP prior, can we show that it fails to capture important aspects?

**Main Review:**

I have a concern with the quality of the paper.

Originality. Even though the CRP is the most well-studied process in Bayesian nonparametrics (BNP), Section 3.1 has found a new use for it, in the context of generating random rectangular partitions. The construction is well-explained by text and illustration (Figure 4, 5, 6). The paper also discusses distinction between the new construction and existing ones where the opportunity arises (lines 166-168). The appearance of the permuton measure gamma creates a degree of flexibility in the prior sample paths (Figure 2) that is worth investigating further.

Quality. (big concern). How is sampling done in the PCRP model? The abstract hints that we will use Markov chain Monte Carlo (MCMC) to draw inference from posteriors in PCRP-based models. For the CRP in clustering models, because of the exchangeability in the partition, we have Gibbs samplers that update the assignment of each observation conditioned on other observations’ assignments. What are the analogous results for PCRP?  I am not sure where the main paper/appendix discusses inference algorithms. While the “Exact BNP model with bridging states” is the closest paragraph, it does not discuss the update steps nor point to a location in the appendix with the equations.

Quality (smaller concern). Is there intuition connecting the density of the permuton to the qualitative behavior in Figure 2? A key appeal of the PCRP approach based on Figure 2 is that different choices of gamma lead to qualitatively different types of rectangular partitioning. Can we understand the connection better? One starting point would be to plot the gamma densities (similar to what the paper currently does in Figure 1e). Currently, the paper points to outside references for information on gamma --- it would be great if it were self-contained. Relatedly, how do the sample paths from the permuton compare to those from say, the Mondrian process, another seemingly flexible prior over rectangular partitions? Finally, in Figure 9, I am not sure what “more detailed clusters” means. Is there some quantitative metric? The observations on the top and bottom rows are not the same as each other as well – for comparisons, it would be better if they were the same.

Clarity. Lines 171-172: what does it mean that the CRP table ordering lead to permutation? What is “n” in this situation? My guess is that n is the number of tables after N customers. If so, I would recommend swapping out n for another notation since having lower case and upper case meaning different objects is confusing. It is not clear what the purpose of the antichain (lines 100) or dashed patterns (lines 103) play in the overall methodology. They are only mentioned in Section 2.1 and not again elsewhere in the paper. In Section 3.2, the intro could be renamed, and the caption of Figure 2 rewritten. The point of Section 3.2 appears to be “for different choices of the permuton gamma, the rectangular partition generated from PCRP takes on one of a few different qualitative behavior.” The starting sentence “how to design the permuton gamma in a specific way” is vague. In Figure 2, the stating sentence should be something like “Five classes of rectangular partitioning corresponding to different choices of permuton gamma.” The current phrasing makes it sound like these are the five classes overall (not just the ones we can generate with the permuton approach). My last comments are small suggestions if time permits. The abstract can be simplified. I think the structure of the abstract in https://www.jmlr.org/papers/volume12/griffiths11a/griffiths11a.pdf is also suitable for your paper. They first state that the sample paths of the IBP are binary matrices; they then mention the type of data analyses in which the IBP would be a good prior. The analogs in your abstract would be lines 6-8 and lines 1-2. In your abstract, the comparison with the BBP/the details on the permuton can be removed, as they introduce a lot of new concepts in the abstract and complicate the main message. The font / tick sizes in Figure 8 should be enlarged. The y-axis can also be standardized across datasets. The plots in Figure 8 are missing ticks on the x-axis. Finally, I would change the tone of “multi-dimensional extension of the CRP … is one of the most critical unsolved issues”, since the word “critical” is very strong and would require a lot of justification. In terms of applied work, the literature has come up with options to model relational data (like the Mondrian process, the IRM etc). In terms of theoretical work, the BBP is already one extension of the CRP in higher dimensions.

Significance. The PCRP has the potential to add to the Bayesian nonparametrics a model over rectangular partitions with more flexible sample paths.

**Time Spent Reviewing:**

3.25

---

> ### Author Response · Authors · 2021-08-10
> **Response to Reviewer rRnG (Part 1/2)**
>
> Once again, we appreciate your very careful reading of our paper and your important comments. In particular, we are delighted to receive your advice on improving the clarity of our paper, which is a role model of how to write a scientific paper. We will first answer the questions you raised and then make specific suggestions for revisions to the manuscript.
>
> > **How is sampling done in the PCRP model? (...) For the CRP in clustering models, because of the exchangeability in the partition, we have Gibbs samplers that update the assignment of each observation conditioned on other observations’ assignments. What are the analogous results for PCRP?**
>
> **PCRP can straightforwardly derive the Gibbs sampling update rule from its generative model, similar to the standard CRP.** First, we would like to give a sketch of the reason for this, and then explain the details. Finally, based on the suggestions we received, we describe how to reorganize the contents of the original paper to improve its clarity for Bayesian inference.
>
> **Sketch** - The key is that if PCRP has $n$ tables during a given iteration of MCMC, all possible rectangular partitions when new table additions occur are at most $(n+1)^2$ patterns. They can be enumerated explicitly using the array representation of random coordinates of PCRP tables. Although there is only one case when a new table occurs in the standard CRP, there are at most $(n+1)^2$ cases when a new table occurs in our PCRP, depending on the rectangular partitions created by the random coordinates of the new table given the current coordinates of $n$ tables. As a result, when we update the table assignment for each row and column of PCRP, we can explicitly enumerate all cases ($n$ cases) of assignments to existing tables and all cases (at most $(n+1)^2$ cases) of assignments to the new table, so we can calculate their posterior probabilities and obtain the Gibbs sampling update rule.
>
> **Details** - We describe how to enumerate at most $(n+1)^2$ rectangular partitions that PCRP can make when it generates a new table. First, for simplicity, we assume that the permuton, which is the prior of the random permutation of PCRP table coordinates, is the uniform distribution on [0,1]x[0,1] (i.e., the uniform permuton), and later we describe how to restrict it to the permutation class (e.g., $2$-clumped permutation) we want. Now we suppose that each of the $n$ tables in PCRP has $n$ coordinates at the $[0,1]\times [0,1]$ plane (Note here that this $[0,1]\times [0,1]$ plane is for representing arrays of permutations (Figure 1 (d)), not for representing rectangular partitions). If we draw crosshairs horizontally and vertically from the coordinates of each table, we can divide the $[0,1]\times [0,1]$ plane into $(n+1)\times (n+1)$ blocks. Now imagine that PCRP generates a new $(n+1)$-th table and tries to specify a random coordinate on the $[0,1]\times [0,1]$ plane. At this time, the new rectangular partition pattern created by the $(n+1)$-table depends only on where the $(n+1)$-th PCRP table falls in the aforementioned $(n+1)\times (n+1)$ blocks of the $[0,1]\times [0,1]$ plane, so there are only a mere $(n+1)^2$ number of cases. Therefore, the calculation of these posterior probabilities can be explicitly obtained. Next, we will describe how to restrict the permutations corresponding to the coordinates in the PCRP table to $2$-clumped permutations. The idea is very simple. For each of the $(n+1)^2$ candidate permutations of the $(n+1)$ PCRP tables, if it does not belong to a $2$-clumped permutation, we simply reject it. If we hope to create a uniform $2$-clumped permutation, we can do the same with minor modifications.
>
> **Improving the clarity** - We will make sure to indicate in the body of the paper where to find the supplementary material on the details of Bayesian inference. As you pointed out, the description of the inference algorithm was very abstract (we focused too much on generalizing the model). We would like to show the whole flow in the form of pseudo-code in terms of practical algorithms, focusing on concrete examples as mentioned earlier. Additionally, we briefly review the development of the MCMC algorithms for conventional relational models, including Gibbs sampling for the infinite relational model [39], the Metropolis-Hastings method [56] and the reversible jump MCMC [Wang et al., SDM2011] for the Mondrian process, and the particle MCMC (sequential Monte Carlo) for the binary space partitioning-tree process [22] and the random tessellation process [31]. Thank you for your helpful comments, which improve the clarity of the manuscript.
>
> [39] C. Kemp, J. B. Tenenbaum, T. L. Griffiths, T. Yamada, N. Ueda, Learning systems of concepts with an infinite relational model, AAAI Conference on Artificial Intelligence (AAAI), pp. 381–388, 2006.
>
> [56] D. M. Roy and Y. W. Teh, The Mondrian process, Advances in Neural Information Processing Systems (NeurIPS), 2009.
>
> [Wang et al., SDM2011] P. Wang, K. B. Laskey, C. Domeniconi, M. I. Jordan, Nonparametric Bayesian co-clustering ensembles. In SDM, pp. 331-342, 2011.
>
> [22] X. Fan, B. Li, and S. A. Sisson, The binary space partitioning-tree process, International Conference on Artificial Intelligence and Statistics (AISTATS), pp. 1859–1867, 2018.
>
> [31] S. Ge, S. Wang, Y. W. Teh, L. Wang, L. Elliott, Random tessellation forests, Advances in Neural Information Processing Systems (NeurIPS), pp. 9575–9585, 2019.
>
> > **Is there intuition connecting the density of the permuton to the qualitative behavior in Figure 2? A key appeal of the PCRP approach based on Figure 2 is that different choices of gamma lead to qualitatively different types of rectangular partitioning. Can we understand the connection better? One starting point would be to plot the gamma densities (similar to what the paper currently does in Figure 1e).**
>
> **Yes, we would like to present the self-similarity property of permuton that may help a little bit with the intuitive understanding of the relationship between permuton and rectangular partitioning.** Additionally, we thank you for your suggestion. As you suggested, we will also depict the permuton corresponding to each rectangular partitioning class to the supplementary material.
>
> **Self-similarity of separable permuton** - Assuming that we have focused on hierarchical and diagonal partitioning, the relationship between permuton and rectangular partitioning becomes clearer. We recall that there is a bijection between hierarchical and diagonal partitioning (i.e., a subset of rectangular partitioning represented by the Mondrian process) and separable permuton (as described in lines 235-238). Surprisingly, the separable permuton has self-similarity [45] (Theorem 1.6 and Figure 2). This gives us the following intuition. Imagine a situation where a separable permuton is viewed as an image, and the resolution is gradually increased from low to high resolution to observe it. This increasing resolution of the separable permuton corresponds to going deeper and deeper in hierarchical rectangular partitioning. In other words, the local detail of the permuton corresponds to the increasing depth of the partition in rectangular partitioning.
>
> We are also aware that several future work remains to be done on the relationship between permuton and rectangular partitioning, including (1) explicit description of the uniform 2-clumped permuton for generic rectangulation (i.e., arbitrary rectangular partitioning), (2) identification of the permutation class that corresponds to hierarchical partitions that do not impose the diagonal requirement (i.e., the class equivalent to the partitions represented by the Mondrian process), and (3) analogy of permuton corresponding to cubic partitions of a high-dimensional array. Research on permuton has been very active in recent years (e.g., [11, 12, 19]), and we hope to see developments in these problems in the near future.
>
> [45] M. Maazoun, On the Brownian separable permuton. Combinatorics, Probability and Computing 29(2), pp. 241–266, 2019.
>
> [11] F. Bassino, M. Bouvel, V. Féray, L. Gerin, and M. Maazoun, A. Pierrot, Universal limits of substitution closed permutation classes, Journal of the European Mathematical Society 22(11), pp. 3565–3639, 2019.
>
> [12] F. Bassino, M. Bouvel, V. Féray, L. Gerin, and A. Pierrot, The Brownian limit of separable permutations, The Annals of Probability 46(4), pp. 2134-2189, 2018.
>
> [19] J. Borga, and M. Maazoun, Scaling and local limits of Baxter permutations and bipolar orientations through coalescent-walk processes, arXiv:2008.09086, 2020.

---

> > ### Author Response · Authors · 2021-08-10
> > **Response to Reviewer rRnG (Part 2/2)**
> >
> > > **Finally, in Figure 9, I am not sure what “more detailed clusters” means. Is there some quantitative metric?**
> >
> > Thank you for your exact point. To clarify our unclear wording, we would like to express our intentions more precisely as follows.
> >
> > - The intention of this part of the explanation was to emphasize the essential difference between the conventional BBP and our proposed CRP.
> >   Specifically, each model differs in its flexibility in finding new block candidates during MCMC iterations: BBP is flexible in inserting a new block in the lower right corner, but operations to modify blocks elsewhere or to add new blocks require long Markov chains that occur with only very small probabilities. PCRP, on the other hand, is able to generate new block insertion candidates everywhere throughout the rectangular partition. As a result, as Figure 9 shows, BBP extracts a very biased rectangular partition due to the fact that the addition of new blocks tends to occur in the lower right corner. In other words, BBP only has a high probability of adding blocks to the lower right, so if a coarse cluster is created in the upper left, it will be difficult to modify the coarse cluster into a fine one. On the other hand, PCRP has the flexibility to add new blocks to the entire rectangular partitioning area, so it can properly find coarse and fine clusters in the entire area. This can be confirmed by the quantitative criterion of prediction performance shown in Figure 8.
> >
> > **[Revision]** Finally, we appreciate your many suggestions for improving the clarity of the manuscript. We feel very fortunate and honored to have received such very enlightening advice as a guide to writing a scientific paper. We know that you have given only a few specific examples of the whole, so we can use your advice to re-examine the whole paper. Here are some specific suggestions for improvement for some of the examples you pointed out.
> >
> > - In Section 2.1 (Preliminaries for permutation), we would like to explain some notions of permutations with concrete examples instead of abstract explanations such as antichain. For example, focusing on 2-clumped permutations, we will provide the following explanation.
> >
> >   "A pair $\sigma_{i}$ and $\sigma_{i+1}$ of a permutation $\sigma$ is a $descent$ of $\sigma$ if $\sigma_{i}>\sigma_{i+1}$. For every descent of $\sigma$, a $clump$ is defined as a maximal set of consecutive values $a,a+1, \dots, b$ with $\sigma_{i}>b>a>\sigma_{i+1}$  such that either all elements of $\{a, a+1,\dots, b\}$ occur to the left of the descent or all elements of $\{a, a+1,\dots, b\}$ occur to the right of the descent. For example, consider a permutation $167439285$. The pair $92$ is just a descent of the permutation $167439285$. Four clumps are associated with this descent, $\{3, 4\}$, $\{5\}$, $\{6, 7\}$, and $\{8\}$. A permutation $\sigma$ is a $k$-clumped permutation if every descent of $\sigma$ has at most $k$ associated clumps. The permutation $167439285$ is $k$-clumped for any $k\geq 4$ because four clumps are associated with the descent $92$, and only shorter clumps are associated with any other descent of the permutation."
> >
> > - In Section 1, Section 3.2, and Figure 2, we would like to adopt your proposal. Thank you very much for the concrete suggestions for revision.
> >
> > - For the abstract, we would like to revise the manuscript as follows. This paper proposes the permuton-induced Chinese restaurant process (PCRP), a stochastic process on rectangular partitioning of a matrix. This distribution is suitable for use as a prior distribution in Bayesian nonparametric stochastic block models for relational analysis to find hidden clusters in matrices and network data.  Our main contribution is to introduce the notion of permutons into the famous Chinese Restaurant Process (CRP) for sequence partitioning: a permuton is a probability measure on $[0, 1]\times [0, 1]$ and can be regarded as a geometric interpretation of the scaling limit of permutations. Specifically, we extend the model that the table order of CRPs has a random geometric arrangement on $[0, 1]\times [0, 1]$ by permuton. Since several important classes of permutons are in one-to-one correspondence with several important classes of rectangular partitions, PCRP can be used as a unified stochastic process to represent several classes of rectangular partitions. This study clarifies the relationship between PCRP and various Bayesian nonparametric rectangular partitions, such as infinite relational models, Mondrian processes, and Block-breaking processes. We also apply PCRP to the Bayesian nonparametric relational analysis of real-world matrices.
> >
> > - For Figure 8 (in Section 5), we will redraw it in a larger font with a tick mark on the x-axis.

---

### Official Review · Reviewer_NeZe · 2021-07-28

**Rating:** 7
**Confidence:** 3

**Summary:**

This paper introduces  a multi-dimensional extension of the Chinese restaurant process (CRP) called permuton-induced CRP (PCRP). The paper discusses the permuton setting of the CRP to get a probabilistic model of rectangular partitioning of a matrix.
A Permuton is a probability measure on [0, 1]x [0, 1], which can be regarded as a geometric interpretation of the
scaling limit of permutations. The PCRP can be used as a unified stochastic process to represent different classes of rectangular partitioning.  The authors also propose a new representation of the BNP model to depict the PCRP as a bridging state to existing BNP relational models.
Experiments involve the application of PCRP to Bayesian nonparametric (BNP) relational data and there are some promising results showcased.

**Ethical Concerns:**

Ethical concerns have been discussed

**Limitations And Societal Impact:**

Yes, limitations have been discussed

**Main Review:**

Originality: The work is a novel combination of well-known techniques. It is clear how this work differs from previous contributions. Related work is adequately cited

Quality: The submission is technically sound justified by theoretical analysis and experimental results. This is a complete piece of work

Clarity: The submission is clearly written.

Significance: The study and results are important. There is potential for others to use the ideas. The submission address a difficult task in a better way than previous work. The work provides a unique theoretical approach.

** Comments
* Can PCRP handle continuous distributions as opposed to a multinomial/Dirichlet distribution?
* How would PCRP behave with an asymmetric Dirichlet prior?
* What are the conditions under which PCRP partitions data to mostly the same partition?

** Minor comments
* Figure 9: It would be helpful if the details of the ‘hidden fine clusters’ can be highlighted. What does ‘hidden’ mean in this context?
* Figure 4, 5 (for example) when mentioned in text: Consider a label for each subfigure (i, ii, or a, b) to increase readability as well as save some space.



**Time Spent Reviewing:**

3

---

> ### Author Response · Authors · 2021-08-10
> **Response to Reviewer NeZe**
>
> We are very encouraged by your very positive comments and glad that you have asked valuable questions about the use of PCRP in machine learning.
>
> > **Can PCRP handle continuous distributions as opposed to a multinomial/Dirichlet distribution?**
>
> **Yes, PCRP can be applied to continuous-valued observation data.** A Bayesian nonparametric clustering for relational data can generally be divided into a prior model for partitioning structure and observation. PCRP only serves as the prior model for partitioning structure and does not impose any additional constraints on the observation part (Typically, for relational models, we often use a model where observations in the same block are sampled from a distribution with the same parameters as the observation part.). For example, a normal-normal model can be used when the observations have continuous values on $\mathbb{R}$, and an exponential-gamma model can be used when the observations have non-negative continuous values on $\mathbb{R}^{+}$.
>
> > **How would PCRP behave with an asymmetric Dirichlet prior?**
>
> We assume your question means the following: how would PCRP behave in an application where observations are categorical values generated from multinomial-asymmetric Dirichlet distribution? As mentioned above, PCRP does not impose any restrictions on the observational data model part, so it is very natural to use the multinomial-asymmetric Dirichlet distribution as the observational data model. We have noticed that asymmetric Dirichlet prior sometimes has a good effect in applications such as topic models (e.g., [Wallach et al., 2009]). By analogy, the asymmetric Dirichlet prior may also be useful in relational data analysis in some applications.  Hopefully, we would have liked to give a concrete example of its use here. However, we could not immediately find any examples of the use of the asymmetric Dirichlet prior in previous studies of Bayesian nonparametric relational model and stochastic block models. We think the points you pointed out are essential to consider in developing the application side of relational data analysis. We are glad for the constructive remarks that will help the community to develop PCRP and relational models in the future.
>
> [Wallach et al., 2009] H.  Wallach, D. Mimno, and A. McCallum, Rethinking LDA: Why Priors Matter, Advances in Neural Information Processing Systems (NeurIPS), 2009.
>
> > **What are the conditions under which PCRP partitions data to mostly the same partition?**
>
> We would like to answer your question with two levels of granularity: (1) What conditions are necessary for different MCMC runs to yield mostly the same results for the same input data, which is desirable in practical use? (2) Whether the posterior distribution of fitting PCRP to the input matrix with potentially infinite size can come close enough to the true rectangular partition when the data has a true rectangular partition? If we have misinterpreted the meaning of your question, we would appreciate it if you could let us know again during the discussion phase.
>
> (1) Local optimum for each MCMC trial - As shown by the standard deviation of perplexity in Figure 8 and Figure 9, we see that PCRP tends to find a different local solution each time for each trial of MCMC with a different random seed. We need to develop a more stable inference algorithm that can find better local optima and we have positive prospects in this regard. **PCRP can potentially lead to collapsed variational Bayesian (VB) inference algorithm for various classes of rectangular partitioning.** In the past, VB algorithms could be constructed only for the regular grid represented by the infinite relational model (e.g., [Ishiguro et al., 2017]), and it was not easy to construct VB for broader classes of rectangular partitions (e.g., hierarchical partitioning of the Mondrian process). PCRP has the potential for breakthroughs in deriving VB algorithms (although, of course, it is not straightforward and requires a bit of ingenuity). We recognize that this is an area for research in the near future.
>
> [Ishiguro et al., 2017] K. Ishiguro, I. Sato, and N. Ueda, Averaged collapsed variational Bayesian inference, Journal of Machine Learning Research, 18(1), pp. 1-29, 2017.
>
> (2) Posterior consistency of PCRP - This is an important question, and we are aware that similar questions have been studied, for example, in the posterior consistency of Dirichlet process mixture models (DPMM) [Ghosh and Ramamoorthi, 2003] [Wu and Ghosal, 2010] and the universal consistency of Mondrian tree/forest [Mourtada et al., 2020]. Intuitively, one would expect that the properties of the standard DPMM would be naturally extended to PCRP, but this is not obvious; we cannot rule out the possibility that the random coordinates of the tables introduced by PCRP do something terrible, and this needs to be carefully investigated. Thank you for this critical question for future research.
>
>  [Ghosh and Ramamoorthi, 2003]  J. K. Ghosh and R. V. Ramamoorthi, Bayesian Nonparametrics, Springer-Verlag, 2003
>
> [Wu and Ghosal, 2010] Y. Wu and S. Ghosal, The L1-consistency of Dirichlet mixtures in multivariate Bayesian density estimation, Journal of Multivariate Analysis 101(10): pp. 2411–2419, 2010.
>
> [Mourtada et al., 2020] J. Mourtada, S. Gaïffas, E. Scornet, Minimax optimal rates for Mondrian trees and forests, Annals of Statistics, 48(4): pp. 2253-2276, 2020.
>
> > **It would be helpful if the details of the ‘hidden fine clusters’ can be highlighted. What does ‘hidden’ mean in this context?**
>
> Thank you for your remarks. To clarify our ambiguous wording, we would like to express our intentions more precisely as follows
>
> - The essential difference between the conventional BBP and our proposed CRP is the flexibility in searching for new block candidates during MCMC iterations: BBP is flexible in inserting a new block in the lower right corner, but operations to modify blocks in other places or add new blocks require long Markov chains that occur with only very small probabilities. PCRP, on the other hand, can generate candidates for new block insertions throughout the rectangular partition. As a result, as Figure 9 shows, BBP has a strong bias towards adopting new block candidates in the lower right corner, which substantially impacts the overall rectangular partitioning. More specifically, once a coarse cluster is created in the upper left corner, it becomes difficult to correct the coarse cluster to a fine cluster by adding blocks to the lower right corner with high probabilities. On the other hand, PCRP is flexible enough to add new blocks throughout the rectangular partitioning, so it can adequately find the coarse and fine clusters throughout. This can be confirmed by the quantitative criteria of the prediction performance shown in Figure 8.

---

### Official Review · Reviewer_hdxZ · 2021-08-01

**Rating:** 7
**Confidence:** 3

**Summary:**

This paper proposes Permuton-induced Chinese Restaurant Processes (PCRPs), a novel class of Bayesian nonparametric (BNP) models for rectangular partitioning. There have been several proposals for BNP priors for rectangular partitioning or more generally space partitioning problems. Many of them can be understood as a multi-dimensional extension of Dirichlet processes, but almost all of them only consider the infinite-dimensional representations (multi-dimensional stick-breaking like representations) which make posterior inference cumbersome. According to the authors, PCRP is probably the first try to introduce a CRP-like representation for rectangular partitioning, which comes with the benefit of handy posterior inference. The main idea is to use some tools developed in the literature dealing with the permutations; it can be shown that a permutation can be mapped to a specific class of rectangular partitioning, and a random permutation can be realized by the random measure called permuton (which is quite similar to graphon). In order to actually draw a rectangular partitioning, a partition is drawn from a plain CRP with table coordinates are sampled from the permuton. Then, the permutation induced from the table coordinates is turned into a rectangular partitioning. PCRP facilitates a convenient MCMC procedure which is demonstrated to converge faster than previous approaches in the experiments but is not projective like normal BNP models. The authors further propose a workaround to this and an MCMC inference for the workaround based on bridging.


**Limitations And Societal Impact:**

The authors admit the theoretical imperfection of the proposed framework and further proposed a workaround for that. The ethical implication is also discussed in section 6.

**Main Review:**

Overall, I find the paper hard to follow. There are many concepts introduced and the paper is too short to faithfully review all those concepts. Personally, I'm not familiar with the concepts related to permutations and patterns, so it took me a while to comprehend the definitions and notations for the permutations and patterns. It is also painful to follow the algorithm; how the permutations are represented as rectangular partitioning and how to convert between those two, and how the diagonal partitionings are extended to generic partitionings.   If possible, it would be better to provide more illustrative examples in the supplementary material for the readers like me. Some notations are used without being introduced; e.g., Av(B) or the symbol $\ngeq$.

I must admit that I failed to understand all the proofs and arguments presented in this paper, so my judgment is for the generic story and implications of the presented work. The paper proposes a novel BNP model for rectangular partitioning based on the connection between permutations and rectangular partitioning. I like the model comes with a practical and possibly easier inference algorithm to implement than the previous ones. I also like to see that the permuton is introduced and led to derive a novel BNP model; this is quite reminiscent of how graphon (graphex) is related to BNP models for graphs and random arrays. The experimental results seem convincing, illustrating the fast convergence of the proposed algorithm as argued in the main paper.

- I wonder about the time complexity of each MCMC step of the proposed algorithm. It seems that whenever a new table is inserted, the permutation structures should be adapted accordingly. How expensive is this procedure, and is there any data structure to efficiently implement and manage this?
- For more generic space-partitioning processes, there have been several works that can partition a space into partitions whose components are not necessarily rectangles. For instance,  Ge et al., 2019 proposes a flexible class of space partitioning process called Random Tessellation Process (RTP, different from Rectangular tilting process) including Mondrian processes as special cases, and include generic non-axis-aligned cut partitions. RTP is also proved to be projective. Of course, the increased flexibility comes at a cost of a complicated inference procedure. I guess probably PCRP is less generic than RTP, but not sure whether the partitions can be generated from PCRP is a subset of partitions that can be generated from RTP. How would you compare PCRP to RTP?
- The perplexity curves in Figure 8 seems not converged for the models other than IRM; If ran long for enough times, do the models converge to similar perplexity values?







**Time Spent Reviewing:**

7 hours

---

> ### Author Response · Authors · 2021-08-10
> **Response to Reviewer hdxZ**
>
> We would like to thank you again for reading our paper so carefully and giving us helpful feedback. First, we will answer the three questions you raised, and then we will show you how to revise our manuscript based on the advice you gave us about improving the clarity of our manuscript.
>
> > **I wonder about the time complexity of each MCMC step of the proposed algorithm. It seems that whenever a new table is inserted, the permutation structures should be adapted accordingly. How expensive is this procedure, and is there any data structure to efficiently implement and manage this?**
>
> **Time complexity for new partition generation** - **This time complexity is $O(n^2)$**, where $n$ is the number of tables of PCRP. We assume that at a certain iteration of MCMC, the PCRP has *n* tables. If the PCRP chooses to add a new table when updating the table assignments for some row or column here, the time complexity of calculating the posterior probabilities of all possible candidates of new rectangular partitions is $O(n^2)$. This time complexity follows immediately from the fact that given $n$ PCRP tables, there are only $(n+1)^2$ possibilities at most for a new rectangular partition. We will explain this in detail below. Each of the *n* tables in PCRP has $n$ coordinates at the $[0,1]\times [0,1]$ plane (Note here that this $[0,1]\times [0,1]$ plane is for representing arrays of permutations (Figure 1 (d)), not for representing rectangular partitions). If we draw crosshairs horizontally and vertically from the coordinates of each table, we can divide the $[0,1]\times [0,1]$ plane into $(n+1)\times (n+1)$ blocks. Now imagine that PCRP generates a new $(n+1)$-th table and tries to specify a random coordinate on the $[0,1]\times [0,1]$ plane. At this time, the new rectangular partition pattern created by the $(n+1)$ PCRP tables depends only on where the $(n+1)$-th PCRP table falls in the aforementioned $(n+1)\times (n+1)$ blocks of the $[0,1]\times [0,1]$ plane, so there are only a mere $(n+1)^2$ number of cases. Therefore, the calculation of these posterior probabilities can be performed in $O(n^2)$.
>
> **Data structure** - We can use an **efficient data structure to represent the generic rectangulation, described in [Merio&Mütze, 2021]** (Section 6.1 and Figure 12). We would like to add a concrete example of this data structure and an updated example of the MCMC step to the supplementary material.
>
> [Merino&Mütze, 2021] A. Merino and T. Mütze, Combinatorial generation via permutation languages. iii. rectangulations, arXiv:2103.09333, 2021 (only referenced in the supplementary material).
>
> > **For more generic space-partitioning processes, there have been several works that can partition a space into partitions whose components are not necessarily rectangles. For instance, Ge et al., 2019 proposes a flexible class of space partitioning process called Random Tessellation Process (RTP, different from Rectangular tilting process) including Mondrian processes as special cases, and include generic non-axis-aligned cut partitions. (...) I guess probably PCRP is less generic than RTP, but not sure whether the partitions can be generated from PCRP is a subset of partitions that can be generated from RTP. How would you compare PCRP to RTP?**
>
> **Partitions drawn from our PCRP are not a subset of partitions drawn from the conventional Random Tessellation Process [31, NeurIPS2019]**. Thank you for the enlightening question. As you pointed out, the models with oblique (sloped) cuts, which have been popular among relational models in recent years, are in a sense more generic than rectangular partitioning models. However, as far as we know, those models can represent only hierarchical partitioning, including the Ostomachion process [27], the binary space partitioning-tree process [22], and the random tessellation process [31]. Therefore, it is not possible to represent a generic rectangulation like the one on the right of Figure 2 with these existing methods. We believe that a model that represents oblique cuts and does not belong to hierarchical partitioning is one of the most important research topics for the future.
>
> [27] X. Fan, B. Li, Y. Wang, and F. Chen, The Ostomachion Process, AAAI Conference on
> Artificial Intelligence (AAAI), pp. 1547–1553, 2016.
>
> [22] X. Fan, B. Li, and S. A. Sisson, The binary space partitioning-tree process, International Conference on Artificial Intelligence and Statistics (AISTATS), pp. 1859–1867, 2018.
>
> [31] S. Ge, S. Wang, Y. W. Teh, L. Wang, L. Elliott, Random tessellation forests, Advances in Neural Information Processing Systems (NeurIPS), pp. 9575–9585, 2019.
>
> > **The perplexity curves in Figure 8 seems not converged for the models other than IRM; If ran long for enough times, do the models converge to similar perplexity values?**
>
> No, we confirmed that even with longer MCMC iterations, PCRP shows a slightly better prediction performance than other methods, i.e., the infinite relational model (IRM), the Mondrian process (MP), and the block-breaking process (BBP). Since BBP [49] and our PCRP have similar expressive power for arbitrary rectangular partitioning, they are theoretically expected to show similar predictive performance if they can iterate MCMC for a nearly infinitely long time. However, in practice, BBP has a strong bias in adding new blocks (as described in lines 210-214 of the text) and is prone to get trapped in local optima, while PCRP can show better performance than BBP. For IRM and MP, they can represent only limited classes of rectangular partitioning. As a result, our proposed CPRP, which can represent a broader class of arbitrary rectangular partitions, showed better prediction performance than IRM and MP. As you point out, Figure 8 does not appear to converge at first glance, but this is mainly because the horizontal axis is logarithmic. Using a logarithmic axis was to convey that PCRP improves the prediction performance from the early stages of the iterations. If we make the horizontal axis linear, we can see that there is sufficient convergence. Thank you for your valuable remarks.
>
> [49] M. Nakano, A. Kimura, T. Yamada, N. Ueda, Baxter permutation process, Advances in Neural Information Processing Systems (NeurIPS), pp. 8648-8659, 2020.
>
> **[Revision]** Thank you for your advice to improve the clarity of this paper. Specifically, we will revise our paper as follows:
>
> - In Section 2.1 (Preliminaries for permutation), we would like to explain the characteristics of each permutation class with concrete examples instead of abstract explanations such as antichain. For example, focusing on 2-clumped permutations, we will provide the following explanation.
>
>   "A pair $\sigma_{i}$ and $\sigma_{i+1}$ of a permutation $\sigma$ is a $descent$ of $\sigma$ if $\sigma_{i}>\sigma_{i+1}$. For every descent of $\sigma$, a $clump$ is defined as a maximal set of consecutive values $a,a+1, \dots, b$ with $\sigma_{i}>b>a>\sigma_{i+1}$  such that either all elements of $\{a, a+1,\dots, b\}$ occur to the left of the descent or all elements of $\{a, a+1,\dots, b\}$ occur to the right of the descent. For example, consider a permutation $167439285$. The pair $92$ is just a descent of the permutation $167439285$. Four clumps are associated with this descent, $\{3, 4\}$, $\{5\}$, $\{6, 7\}$, and $\{8\}$. A permutation $\sigma$ is a $k$-clumped permutation if every descent of $\sigma$ has at most $k$ associated clumps. The permutation $167439285$ is $k$-clumped for any $k\geq 4$ because four clumps are associated with the descent $92$, and only shorter clumps are associated with any other descent of the permutation."
>
> - In Section 3.1 (Model description of our PCRP), we would like to improve the explanation of the model by making it more intuitive and easier to understand.  In Figure 5, the concept of stair-stepping nature will be written directly in the diagram to explain the transformation from permutation to diagonal rectangulation.  For the transformation from diagonal rectangulation to generic rectangulation in Figure 6, we would like to add the explicit part of the permutation where the transformation takes place in the diagram. Moreover, in the supplementary material, we show another example for these transformations in the case of a smaller permutation.

---

> > ### Comment · Reviewer_hdxZ · 2021-08-24
> > **Thanks**
> >
> > I appreciate author's response which resolved most of the questions I raised. I keep my score intact.

---

### Official Review · Reviewer_sbRq · 2021-08-03

**Rating:** 8
**Confidence:** 4

**Summary:**

This paper extends the traditional ONE-DIMENSIONAL Chinese Restaurant Process into the multi-dimensional case and introduce the permutation into the Bayesian nonparametric clustering perspective. As it can be seen as an component-wise integrated-out Block-Breaking Process, this work is very innovative and should produce long-term impact to the Bayesian nonparametric community.

**Main Review:**

This paper presents a very innovative approach in extending the traditional one-dimensional CRP into the multi-dimensional case. Instead of independently and separately regarding the multi-dimensional clustering through multiple CRPs, this approach combines the multi-dimensional clustering into one single approach and is a very neat solution. I would very much recommend it to be accepted.

The presentation is very clear and the figure visualisation helps a lot in understanding the related processes.

I do not have suggestions to further improve the paper (as the paper is already nearly perfect). It would be great if the authors can answer the below two small questions:
1, can the model extends to 3-dimensions or more. The current work seems to mainly focus on the 2-dimensional case. I believe this approach should be able to work on 3 dimensions or more in theory. However, will the related process change a lot and need a lot more considerations or more unseen difficulties?

2, recently, there is another paper of Hierarchical Infinite Relational Model (UAI-2021). That paper should be different from this approach. Can the authors comment on that paper?

**Time Spent Reviewing:**

2 hours

---

> ### Author Response · Authors · 2021-08-10
> **Response to Reviewer sbRq**
>
> We are very encouraged by your positive feedback and are glad to respond to your questions about the future development of this field.
>
> > **Can the model extends to 3-dimensions or more?**
>
> To our current knowledge, **the answer is partly yes and partly no**. Specifically, in high-dimensional cases, we find it challenging to apply PCRP to arbitrary partitioning, while at least we can apply it to hierarchical partitioning. If we only focus on the hierarchical partitions (i.e., guillotine partitions) of higher-dimensional arrays, then PCRP can be extended to a model that represents them. This can be done based on the following fact known in combinatorics.
>
> [*, Theorem 5] There exists a bijection between the set of guillotine partitions of a $2^{d-1}$-dimensional array by $n$ cuts and the set of separable $d$-permutations of $\{1,\dots, n+1\} $.
>
> This fact implies that we can represent guillotine partitions of a $2^{d-1}$-dimensional array by constructing a model using $d$ separate PCRPs (with separable permutons). However, it is not straightforward to extend this result to a broader class of cubic partitions (e.g., a higher-dimensional analogy of generic rectangulations). The reason for this is that we have not yet found a combinatorial object that has a one-to-one correspondence for arbitrary cubic partitions of an array of three or more dimensions. Thank you for the important inquiry that will lead to the future development of this field.
>
> [*] A. Asinowski and T. Mansour, Separable $d$-permutations and Guillotine partitions, Annals of Combinatorics, 14(1): 17-43, 2010.
>
> > **There is another paper on Hierarchical Infinite Relational Model (HIRM) (UAI-2021). That paper should be different from this approach. Can the authors comment on that paper?**
>
> We also recently came across this UAI paper and are glad to see the development of related fields. In short, **we see the HIRM as a near-future collaborator rather than a competitor to our PCRP**. From our point of view, HIRM is expected to be more attractive when applied to high-dimensional arrays (as you mentioned in your first question). The key of HIRM is that it defines a probability distribution over relation clusters, domain entity clusters, and relation values, which is a new perspective on modeling relational data. As mentioned by the authors of the UAI paper in the fourth paragraph of Section 6, using the Mondrian process as an example, HIRM can be combined with conventional relational models other than IRM. Therefore, our PCRP should be able to be extended to extract relational clusters and domain entity clusters by combining with HIRM ideas. We are happy to see the new ideas of HIRM as a tool to make various relational models, including PCRP, more effective in real-world data science applications.

---

### Decision · Program_Chairs · 2021-09-27

**Decision:**

Accept (Poster)

**Comment:**

The reviewers all agreed that this paper should be accepted. Please read through the reviews and responses and make sure to include all suggested changes in the camera ready version. One area to pay special attention to when preparing the camera ready is clarity. For example, this paper contains quite technical definitions of permutations/partitions and their relationships. It would be very useful to the more general reader to provide many more illustrative examples (in the appendix, if space is limiting).

As a side note, the reviewers greatly appreciated the authors' enthusiastic engagement during the discussion period -- well done!